# Cholesterol metabolism drives regulatory B cell IL-10 through provision of geranylgeranyl pyrophosphate

Jack A. Bibby [1,2✉], Harriet A. Purvis [1], Thomas Hayday[1], Anita Chandra [3], Klaus Okkenhaug [3], Sofia Rosenzweig [4], Ivona Aksentijevich[4], Michael Wood[5], Helen J. Lachmann[5], Claudia Kemper [1,2,6], Andrew P. Cope [1,7,8✉] & Esperanza Perucha [1,7,8✉]

Regulatory B cells restrict immune and inflammatory responses across a number of contexts. This capacity is mediated primarily through the production of IL-10. Here we demonstrate that the induction of a regulatory program in human B cells is dependent on a metabolic priming event driven by cholesterol metabolism. Synthesis of the metabolic intermediate geranylgeranyl pyrophosphate (GGPP) is required to specifically drive IL-10 production, and to attenuate Th1 responses. Furthermore, GGPP-dependent protein modifications control signaling through PI3Kδ-AKT-GSK3, which in turn promote BLIMP1-dependent IL-10 production. Inherited gene mutations in cholesterol metabolism result in a severe autoinflammatory syndrome termed mevalonate kinase deficiency (MKD). Consistent with our findings, B cells from MKD patients induce poor IL-10 responses and are functionally impaired. Moreover, metabolic supplementation with GGPP is able to reverse this defect. Collectively, our data define cholesterol metabolism as an integral metabolic pathway for the optimal functioning of human IL-10 producing regulatory B cells.

[1] Centre for Inflammation Biology and Cancer Immunology, School of Immunology and Microbial Sciences, King's College London, London SE1 1UL, UK. [2] Complement and Inflammation Research Section (CIRS), National Heart, Lung, and Blood Institute, National Institutes of Health, Bethesda, MD 20892, USA. [3] Division of Immunology, Department of Pathology, University of Cambridge, Cambridge CB2 1QP, UK. [4] Inflammatory Disease Section, National Human Genome Research Institute, National Institutes of Health, Bethesda, MD 20892, USA. [5] National Amyloidosis Centre, Division of Medicine, University College London and Royal Free Hospital London NHS Foundation Trust, London NW3 2PF, UK. [6] Institute for Systemic Inflammation Research, University of Lübeck, Lübeck, Germany. [7] Centre for Rheumatic Diseases, King's College London, London SE1 1UL, UK. [8] These authors contributed equally: Andrew P. Cope, Esperanza Perucha. ✉email: jack.bibby@nih.gov; andrew.cope@kcl.ac.uk; esperanza.perucha@kcl.ac.uk

Immunosuppressive B cells form a critical component of the immune regulatory compartment[1,2]. It is thought that their suppressive capacity derives mainly from their ability to produce IL-10, and in the absence of any lineage marker, this is considered a hallmark of regulatory B cells[3-6]. Their functional importance has been well described in mouse models of disease, demonstrating a potent regulatory capacity across a number of contexts including infection, cancer, and autoimmune disease[3,7-10]. Comparable suppressive activity has been demonstrated in human regulatory B cells in vitro, suggesting that these cells also contribute toward regulating inflammatory responses in humans[5,6]. In support of this, IL-10 producing B cells have been demonstrated to be numerically depleted or functional impaired, ex vivo, in patients with inflammatory disease[5,11,12].

Despite the importance of IL-10 production by B cells, relatively little is known about the molecular mechanisms that govern its expression. Typically, induction of a regulatory phenotype in both human and mouse B cells has been achieved through ligation of TLR9 or CD40[5,6,13]. Downstream, PI3K and ERK signals appear to be important for IL-10 expression[14,15]. This is in broad agreement with in vivo data from mouse models suggesting that both TLR and CD40 signals are required for the induction of IL-10 in response to inflammatory stimuli[3,13]. However, precise details of the signaling cascades or cellular profiles underpinning the induction of a regulatory program in B cells are poorly understood.

Control of immune cell metabolism is critical in regulating fundamental immunological processes[16,17]. However, in comparison to innate cell and T-cell lineages, there has been relatively little work aimed at understanding the metabolic regulation of B-cell biology[18-20]. Furthermore, there is currently no understanding toward the metabolic requirements of regulatory B cells. An emerging concept from studies of regulatory T cells and M2 macrophages is their heightened reliance on lipid metabolism in comparison to the metabolic requirements of inflammatory immune cell lineages. Much of this work has focused on fatty acid oxidation, defining this as a central pathway that underpins polarization and effector functions in regulatory cells[21]. In contrast, the contribution of cholesterol metabolism (the multistep conversion of HMG-CoA to cholesterol) to immune function is less well characterized. Our recent studies, as well as those of others, suggest that cholesterol metabolism plays a role in restricting inflammatory responses[22-24]. These data are consistent with patients carrying mutations in the pathway, who develop severe and recurring autoinflammatory fevers, associated with dysregulated B-cell responses, manifest by elevated serum immunoglobulin levels[25].

Given the immunoregulatory associations with cholesterol metabolism, we set out to address its role in IL-10 producing B cells. In doing so, we reveal that cholesterol metabolism is a central metabolic pathway required for the induction of a regulatory phenotype in human B cells, and define the specific metabolic intermediate of the pathway that mediates critical signaling events for the induction of a regulatory B-cell program. In vivo evidence for this specific metabolic requirement is provided by virtue of defective regulatory responses of B cells derived from patients with inherited defects in cholesterol metabolism.

## Results

**Cholesterol metabolism drives regulatory B cell function**. Given that patients with dysregulated cholesterol metabolism generate severe systemic inflammation and hyperactive B-cell responses, we investigated the role of cholesterol metabolism in regulatory B-cell function (the metabolic pathway is depicted in Supplementary Fig. 1). Consistent with previous observations, TLR9 ligation in human B cells induces a potent IL-10 response (Fig. 1a, b, Supplementary Fig. 2a, b), which mediates their regulatory phenotype[5,11]. In agreement, TLR9 activated B cells were able to suppress Th1 induction in CD4+ T cells in vitro (protocol outlined in Fig. 1c). Neutralization of IL-10 resulted in a complete loss of suppression, confirming the dependence on IL-10 in mediating this function (Fig. 1d). In line with this, concentrations of IL-10 produced by B cells upon TLR9 stimulation (>1 ng ml$^{-1}$, Fig. 1b) were sufficient to directly suppress CD4+ T cell IFNγ expression (Supplementary Fig. 2c). In contrast, pre-treatment of B cells with the HMG-CoA reductase inhibitor atorvastatin, which inhibits an early step of the cholesterol metabolic pathway, abrogated this suppressive capacity. Furthermore, supplementation of exogenous mevalonate reversed this defect, suggesting the requirement for a specific metabolite downstream of HMG-CoA (Fig. 1d).

The loss of suppressive capacity by IL-10 neutralization and HMG-CoA reductase inhibition indicated that cholesterol metabolism could be directly regulating IL-10 production. Indeed, inhibition of HMG-CoA reductase impaired the ability of B cells to produce IL-10, both at the mRNA and protein level (Fig. 1e, f, Supplementary Fig. 3a, b). Once again, and consistent with metabolic depletion downstream of HMG-CoA, exogenous mevalonate was able to reverse this defect (Fig. 1e, f). In contrast to IL-10, we observed a concurrent preservation in the inflammatory response, as determined by TNF and IFNγ protein, and *IL6*, and *LTA* gene expression (Fig. 1g, h, Supplementary Fig. 3c, d). Together, these data indicated that cholesterol metabolism was critical in mediating IL-10 expression, and therefore the anti-inflammatory function of human B cells.

**Cholesterol metabolism drives IL-10 independent of phenotype**. We next aimed to understand how cholesterol metabolism was able to mediate IL-10 production. Certain populations of human B cells have been proposed as primary producers of IL-10. The most well characterized of these are CD24$^{hi}$CD27$^+$ (B10) and CD24$^{hi}$CD38$^{hi}$ B cells[5,6]. In agreement with previous observations, we observed that all populations measured (B10, CD24$^{hi}$CD38$^{hi}$, naïve, memory, and plasmablast) contribute to the pool of IL-10 expressing cells to varying degrees after stimulation with CpG (IL-10$^+$ cells ranging from 1 to 12% of B-cell populations, Supplementary Fig. 4a, b). Furthermore, B10 and CD24$^{hi}$CD38$^{hi}$ B cells produced higher levels (two to threefold) of IL-10 in response to TLR9 stimulation (Supplementary Fig. 4b). Acquiring the capacity to produce IL-10 showed no dependence on proliferation, as IL-10 production was seen irrespective of proliferation state (Supplementary Fig. 4c). Following inhibition of HMG-CoA reductase we observed no change in frequencies of B cell populations, viability, or cell surface markers (HLA-DR, CD86, or CD40), excluding the possibility that perturbation of cholesterol metabolism was depleting specific B-cell subsets that possess a greater propensity to express IL-10 (Supplementary Fig. 4d–f). Furthermore, HMG-CoA reductase inhibition resulted in a 2–3-fold reduction in IL-10 expression irrespective of B-cell population (either naïve, memory, B10, or CD24$^{hi}$CD38$^{hi}$, Supplementary Fig. 4g). Therefore, these data indicated a role for cholesterol metabolism in regulating IL-10 production that is shared across B-cell populations, rather than an effect on specific populations.

**Cholesterol metabolism drives IL-10 via GGPP**. To more precisely understand the mechanistic control by cholesterol metabolism, we next sought to investigate if a specific pathway metabolite downstream of HMG-CoA was regulating IL-10. Cholesterol metabolism encompasses a number of metabolic

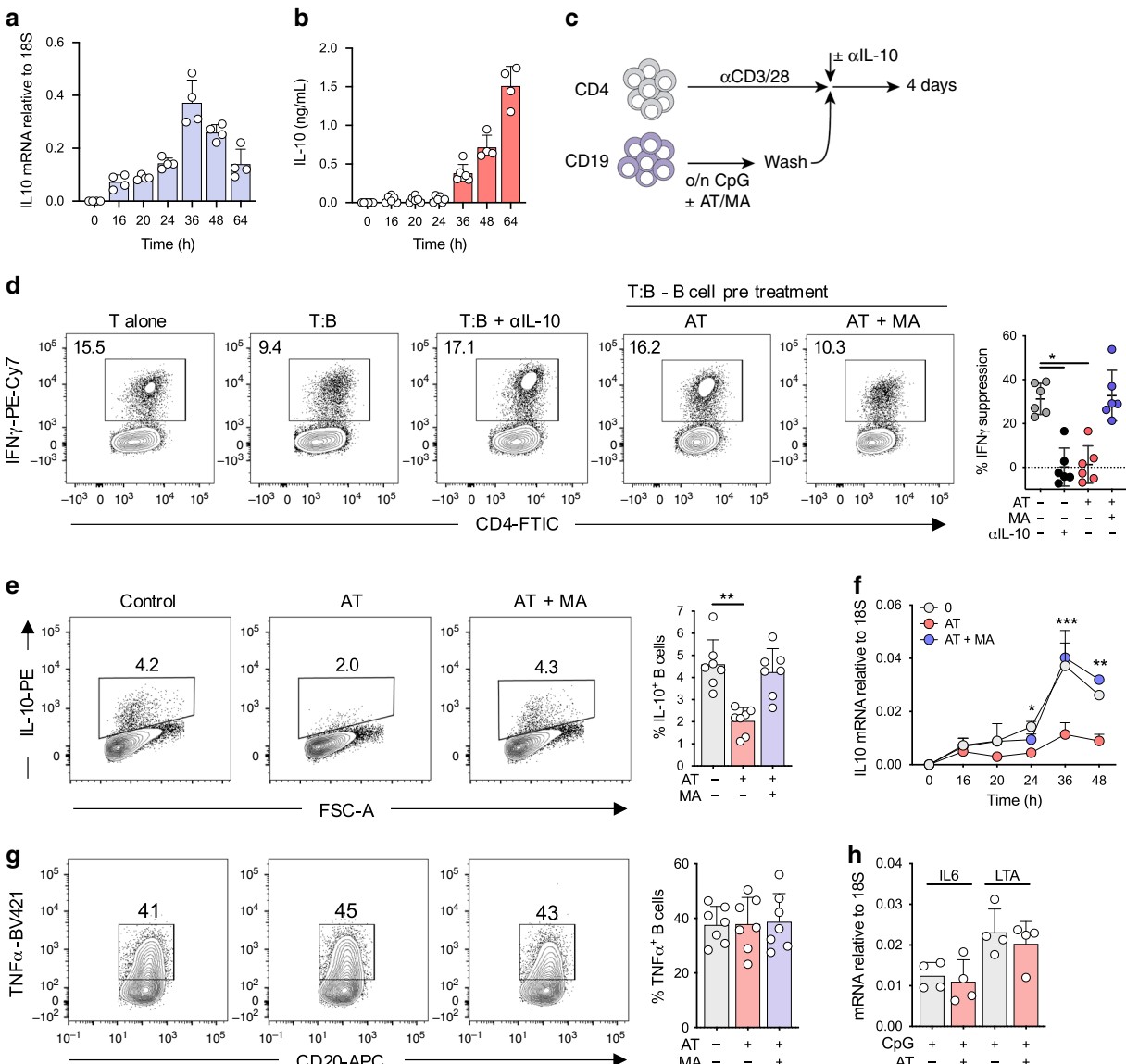

**Fig. 1 Cholesterol metabolism drives regulatory B cell function. a, b** Kinetics of *IL10* mRNA transcript (**a**) and IL-10-secreted protein (**b**) expression at various time points in human B cells after TLR9 stimulation ($n = 4$ for mRNA or $n = 4$–5 for protein). *IL10* mRNA was measured by qRT-PCR, and calculated relative to *18S*. **c** Schematic protocol for the co-culture for T and B cells. Briefly, human B cells were stimulated through TLR9 ± atorvastatin (AT) ± mevalonate (MA) overnight, before washing and addition to anti-CD3/28 stimulated human CD4+ T cells for 4 days ± αIL-10 antibody. **d** IFNγ production in human CD4+ T cells after co-culture with autologous TLR9-activated B cells ($n = 6$, pval = 0.03 and 0.02). IFNγ suppression was calculated by $((x - y)/x)*100$ where $x$ = IFNγ production by T cells alone, $y$ = IFNγ production in co-cultured T cells. **e** IL-10 expression in human B cells after stimulation through TLR9 ± AT ± MA ($n = 7$, pval = 0.003). **f** IL-10 mRNA expression over time in human B cells after stimulation through TLR9 ± AT ± MA ($n = 4$). **g** TNF expression in human B cells after stimulation through TLR9 ± AT ± MA ($n = 7$). **h** *IL6* or *LTA* mRNA expression, relative to *18S*, in human B cells after stimulation through TLR9 ± AT ($n = 4$). Each data point represents individual donors. All data presented are mean ± SD. Statistical testing in all figures was done by a Friedman's test with Dunns's multiple comparisons test, or for (**f**) a mixed-effects model with Dunnett's multiple comparisons test. *$P < 0.05$, **$P < 0.01$, ***$P < 0.001$, and all significant values are shown.

pathways implicated in immune function including mevalonate, isoprenyl and sterol metabolism (Supplementary Fig. 1), all of which are attenuated by HMG-CoA reductase inhibition to varying degrees. Given that defects in the isoprenyl branch have been demonstrated to result in hyperinflammatory responses in vivo[23,26], we investigated if isoprenylation was regulating IL-10. To this end, we targeted geranylgeranyltransferase (GGTase) and farnesyltransferase (FTase), enzymes known to post-translationally modify proteins with geranylgeranyl pyrophosphate (GGPP) or farnesyl pyrophosphate (FPP) groups respectively (enzymes and metabolites outlined in Fig. 2a). Inhibition of

GGTase, but not FTase, substantially reduced TLR9-induced IL-10 production, indicating that geranylgeranyl dependent modifications regulate IL-10 expression (Fig. 2b, Supplementary Fig. 5a, b). In keeping with the effects of HMG-CoA reductase inhibition, inflammatory cytokine production was preserved (Fig. 2c). Furthermore, we observed no or little effect on the proliferation, differentiation, and antibody production by B cells after TLR9 ligation in the presence of either atorvastatin or GGTi during longer cultures (5–7 days, Supplementary Fig. 5c). In experiments to test GGTase specificity, we also observed that IL-10 was dependent on GGTase-I, but not GGTase-II, as inhibition

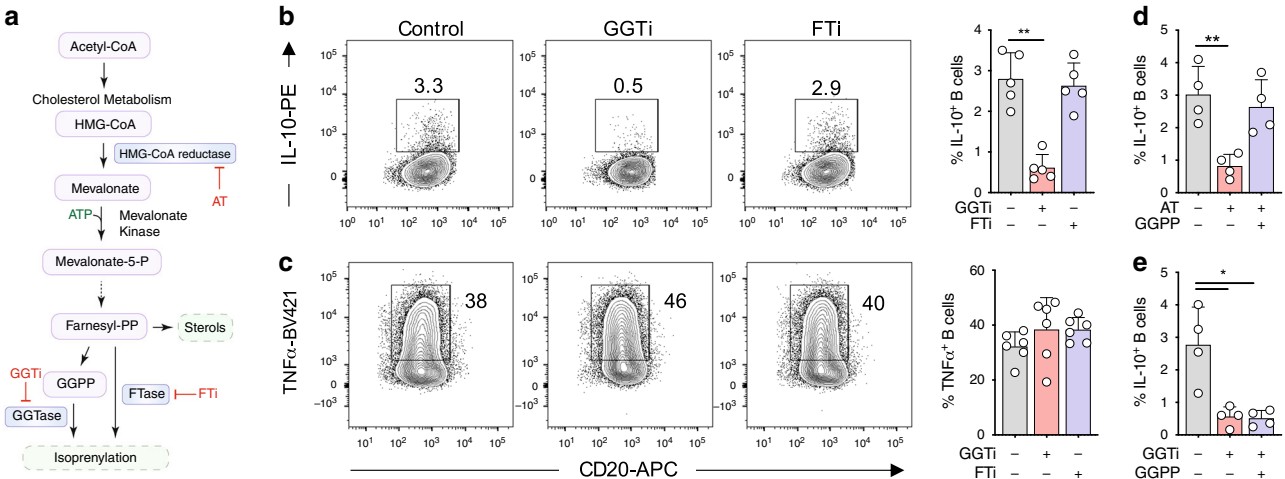

**Fig. 2 Cholesterol metabolism drives IL-10 via GGPP. a** Schematic diagram showing key metabolites and enzymes of the isoprenylation route in cholesterol metabolism. **b**, **c** IL-10 (**b**) and TNF (**c**) expression in human B cells after stimulation through TLR9 ± geranylgeranyl transferase inhibition (GGTi, GGTi-298) ± farnesyl transferase inhibition (FTi, FTi-277) (*n* = 5 for IL-10, pval = 0.001; *n* = 6 for TNF). **d** IL-10 expression in human B cells after stimulation through TLR9 ± atorvastatin (AT) ± geranylgeranyl pyrophosphate (GGPP) (*n* = 4, pval = 0.009). **e** IL-10 expression in human B cells after stimulation through TLR9 ± GGTi ± GGPP (*n* = 4, pval both 0.03). Each data point represents individual donors. All data presented are mean ± SD. Statistical testing in all figures was done by a Friedman's test with Dunns's multiple comparisons test. *$P < 0.05$, **$P < 0.01$, and all significant values are shown.

of GGTase-II upon TLR9 ligation did not affect IL-10 expression (Supplementary Fig. 5d).

To support the notion of GGPP dependency, we depleted metabolites within the pathway with atorvastatin, and found that specifically supplementing the geranyl branch with exogenous GGPP prevented the blockade of IL-10 production (Fig. 2d, Supplementary Fig. 5e). In agreement with a specific effect of GGPP, supplementation with squalene—a metabolite downstream of the isoprenylation branch—was unable to rescue IL-10 production (Supplementary Fig. 5f). Finally, to understand if cellular utilization of GGPP was dependent on the enzymatic activity of GGTase, we supplied cells with exogenous GGPP together with inhibition of GGTase. We found that GGPP was unable to rescue levels of IL-10 (Fig. 2e), consistent with the idea that both GGTase activity and GGPP sufficiency are required for TLR9 induced expression of IL-10.

**GGPP regulates signaling through TLR9.** The most well characterized targets of isoprenylation are Ras superfamily proteins[27,28]. Ras-dependent pathways downstream of TLR9 include Raf-MEK-ERK and PI3K-AKT cascades (Fig. 3a). Signaling through both pathways is critically required for IL-10 production, as inhibition of either pathway is sufficient to block TLR9-dependent IL-10 induction (Fig. 3b, c). Therefore, we sought to address if activation of these pathways is dependent on GGPP. Following GGTase inhibition, AKT phosphorylation on Ser473 was severely impaired, whereas ERK phosphorylation was modestly reduced at early timepoints (Fig. 3d, quantification shown in Supplementary Fig. 6a, b). Downstream, AKT restricts GSK3 activity through an inhibitory phosphorylation event targeting Ser9[29]. In keeping with the idea that GSK3 suppression is required for IL-10 production, chemical inhibition of GSK3 enhanced TLR9-induced IL-10 expression (Fig. 3e). Blocking GGTase activity resulted in reduced Ser9 phosphorylation on GSK3. This indicated preserved activation of GSK3, and that GGTase activity negatively regulates GSK3, which in turn is necessary for IL-10 production (Fig. 3f). We then examined if inhibition of GSK3 was sufficient to rescue an upstream perturbation of geranylgeranylation. Indeed, bypassing GGTase through

GSK3 inhibition was able to fully rescue IL-10 expression, without affecting TNF (Fig. 3g, Supplementary Fig. 6c). Together these data suggest that following TLR9 engagement, IL-10 induction through AKT-GSK3, and possibly ERK, is dependent on GGTase activity.

**PI3Kδ regulates IL-10 expression in human and mouse B cells.** We next aimed to determine how geranylgeranyl-driven phosphorylation cascades regulate AKT-GSK3 signaling. PI3K mediates Ras-dependent AKT signaling, which suggests that isoprenylation-driven phosphorylation cascades through AKT are dependent on PI3K activity. Accordingly, pan inhibition of PI3K blocked expression of IL-10 upon TLR9 stimulation (Fig. 4a). We found, by selective inhibition of either δ, α, or γ isoforms of PI3K, that IL-10 is primarily regulated through PI3Kδ downstream of TLR9 (Fig. 4b). To further support the importance of PI3Kδ, we examined IL-10 production in splenic B cells derived from mice expressing either a hyperactive (E1020K) or catalytically inactive (D910A) PI3Kδ subunit. Following TLR9 activation, presence of the hyperactive PI3Kδ mutant resulted in threefold increased expression of IL-10, whereas the catalytically inactive PI3Kδ resulted in almost a complete loss of IL-10 production (Fig. 4c). Furthermore, and in agreement with our previous observations, TNF production was preserved (Fig. 4c). These data suggest that isoprenylation-dependent interactions between Ras and PI3Kδ are required for IL-10 production, and likely underpin the regulatory function of B cells.

**GGPP regulates IL-10 induction via BLIMP1.** Currently there is no defined transcription factor that regulates IL-10 in human B cells. Therefore, we sought to understand how IL-10 is transcriptionally regulated, and to clarify the role for GGPP in this process. Stimulation of B cells in the presence of actinomycin D indicated that a transcriptional event within the first 24 h was necessary for IL-10 production (Supplementary Fig. 7a). Therefore, we suspected that under conditions of GGTase inhibition, expression of a transcription factor necessary for IL-10 may be compromised. To test this, we performed RNA-sequencing on TLR9-stimulated human B cells in the presence or absence of

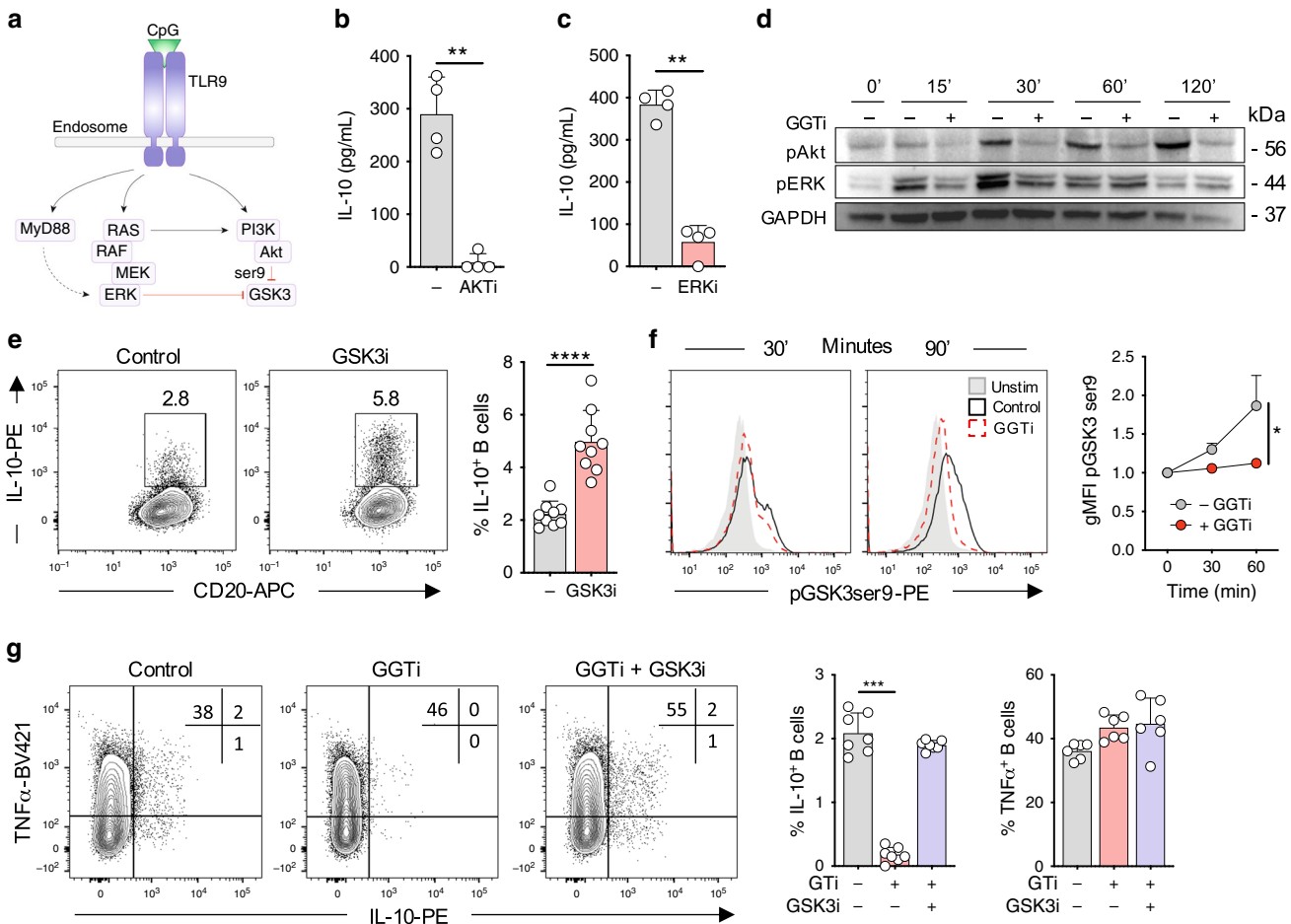

**Fig. 3 GGPP regulates signaling through TLR9. a** Schematic of Ras-dependent signaling cascades downstream of TLR9. **b**, **c** IL-10 expression in human B cells stimulated through TLR9 ± inhibition of (**b**) AKT (AKTi, MK-2206) or (**c**) ERK (ERKi, SCH772984) ($n = 4$, pval = 0.005, and 0.001). **d** Phosphorylation of AKT (ser473) or ERK1/2 (Thr202/Tyr204) over time in human B cells upon stimulation through TLR9± inhibition of geranylgeranyl transferase (GGTi) (representative of $n = 3$). **e** IL-10 expression in human B cells stimulated through TLR9± inhibition of GSK3 (GSK3i, CHIR-99021) ($n = 9$, pval < 0.001). **f** Phosphorylation of GSK3 (ser9) over time in human B cells upon stimulation through TLR9± inhibition of geranylgeranyl transferase (GGTi-298), relative to unstimulated controls ($n = 4$, pval = 0.02). **g** IL-10 and TNF expression in human B cells stimulation through TLR9 ± GGTi ± GSK3i ($n = 7$, pval = 0.0001). Each data point represents individual donors. All data presented are mean ± SD. Statistical testing in figures in (**b**), (**c**), and (**e**) was done by a paired $t$ test, **f** using a twoway ANOVA with Sidak's multiple comparisons test, and **g** by Friedman's test with Dunns's multiple comparisons test **\*\*$P < 0.01$, \*\*\*$P < 0.001$, \*\*\*\*$P < 0.0001$ and all significant values are shown.

GGTase inhibition. Principal component analysis demonstrated that the main variation in the transcriptional profile of the cells was dependent on GGTase activity, as expected (Fig. 5a, Supplementary Fig. 7b). We then interrogated the list of differentially expressed genes (±1.5-fold, false-discovery rate (FDR) < 0.05) for those known or likely to encode human transcription factors (as defined by ref. [30]). Among the 73 differentially expressed transcription factors, 9 have been shown to regulate IL-10 expression in other cell types (Fig. 5b–d, Supplementary Fig. 7c), either through activation or repression (defined in ref. [31]). We decided to focus on *BLIMP1* for two reasons. First, its transcript is enriched in IL-10[+] human B cells[32]. Secondly, interrogation of a mouse B cell ChIP- and RNA-seq data set[33] demonstrated that binding of BLIMP1 to the *Il10* locus correlates with increased IL-10 expression (Supplementary Fig. 7d). We first confirmed reduced expression of BLIMP1 protein upon GGTase inhibition (Supplementary Fig. 7e), and consistent with previous data, we observed increased expression of BLIMP1 protein within the IL-10[+] B cell population (Fig. 5e). To more thoroughly address its role, we performed gene targeting experiments on BLIMP1 in primary human B cells. TLR9 stimulation strongly upregulated

BLIMP1 expression in multiple B-cell populations, including, unexpectedly, recently activated naïve and CD24[hi]CD38[hi] B cells (Fig. 5f, Supplementary Fig. 7f, g), while siRNA knockdown was able to reduce TLR9-dependent protein level expression by ~60% (Fig. 5g, h). In agreement with a central role in regulating IL-10, siRNA-mediated knockdown of BLIMP1 reduced IL-10 expression by 50–90%, whereas TNF production was preserved (Fig. 5i). Together, these data demonstrate that BLIMP1 regulates IL-10 in human B cells, and its expression is dependent on cholesterol metabolism for its provision of GGPP.

**MKD patients generate poor regulatory B-cell responses**. Our data demonstrated that cholesterol metabolism is essential for B cell derived IL-10 production. We therefore sought to validate our findings in the context of human inflammatory disease. To address this, we studied MKD patients, who carry a partial loss-of-function mutation in the mevalonate kinase gene, whose cognate substrate lies directly downstream of HMG-CoA in the metabolic pathway (highlighted in Supplementary Fig. 1a). These patients present with a complex and severe autoinflammatory syndrome[25]. We first compared peripheral regulatory B cells from

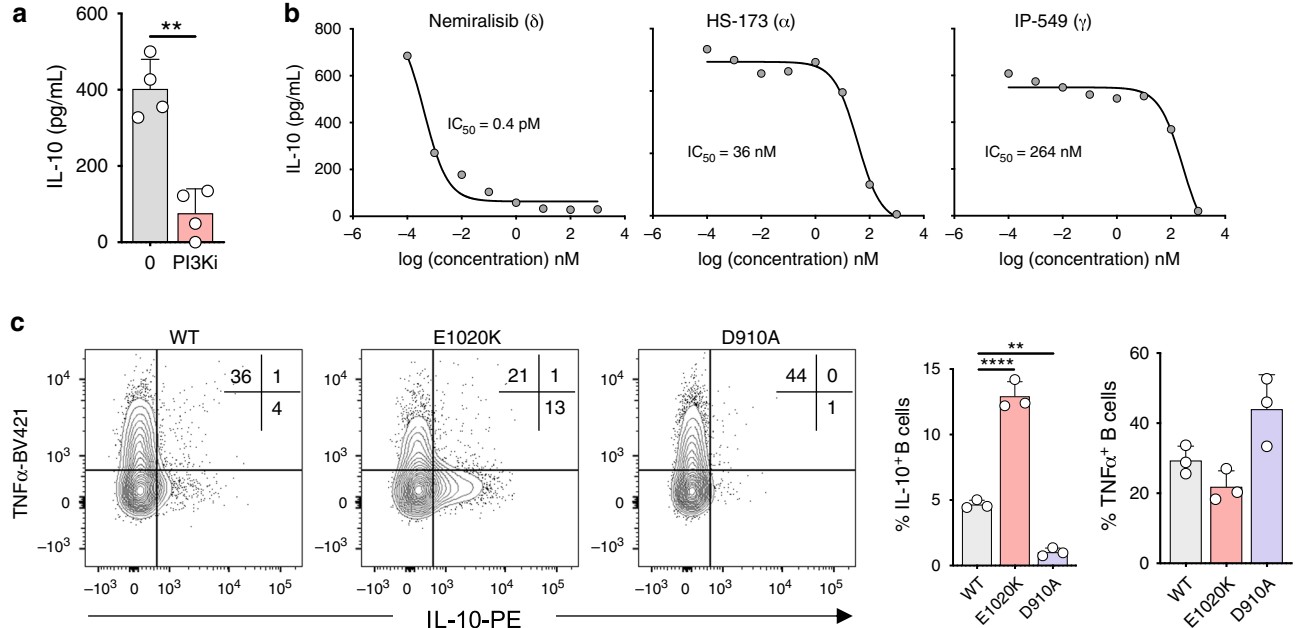

**Fig. 4 PI3Kδ regulates IL-10 expression in human and mouse B cells. a** IL-10 expression in human B cells stimulated through TLR9± pan inhibition of PI3K (PI3Ki, ZSTK474) ($n = 4$, pval = 0.008). **b** IL-10 expression in human B cells stimulated through TLR9± inhibitors for specific family members of PI3K: these being δ (Nemiralisib), γ (IP-549), or α (HS-173), over a range of concentrations ($n = 4$). **c** IL-10 and TNF expression in mouse B cells after stimulation through TLR9. Mice used were wild type (WT), hyperactive PI3Kδ (E1020K), or catalytically inactive PI3Kδ (D910A) ($n = 3$, pval = 0.002 and <0.0001). Each data point represents individual donors or mice. Data presented are mean ± SD. Statistical testing in figures in **a** was done by a paired $t$ test, or in **c** by a one-way ANOVA with Dunnet's multiple comparisons test. **$P < 0.01$, ****$P < 0.0001$ and all significant values are shown.

MKD patients to age and sex matched healthy controls (Patient information in Supplementary Fig. 8a). We observed similar frequencies of B10 cells, whereas $CD24^{hi}CD38^{hi}$ B cells were reduced (Fig. 6a, b). This is in line with a general shift from a naïve to memory like B cell phenotype in MKD patients (Supplementary Fig. 8b). In longer term in vitro cultures, we observed that B cells from MKD patients possess a normal capacity to proliferate, differentiate, and to produce antibodies (Supplementary Fig. 8c).

We next examined if dysregulated cholesterol metabolism in patients with MKD leads to an inability to mount an IL-10 response, which could contribute to disease persistence or exacerbation. In line with this hypothesis, MKD patients generated poor IL-10 responses after stimulation through TLR9 (30–70% reduction versus controls, Fig. 6c), while TNF expression was enhanced in 2 of 4 donors (Supplementary Fig. 8d). Interestingly, the defect in IL-10 production could be reversed by the addition of GGPP, indicating that at least in part, this defect was due to a relative deficiency of the isoprenyl metabolite (Fig. 6c). As with our previous observations, IL-10 expression was reduced across all B-cell phenotypes, including an approximately threefold reduction within B10 and $CD24^{hi}CD38^{hi}$ B cells (Fig. 6d, gating in Supplementary Fig. 4a, d).

We previously identified GSK3 as a key node in IL-10 production (Fig. 3e–g). Therefore, we addressed if reduced IL-10 levels in MKD patients could be rescued through inhibition of GSK3, which would provide a rescue to signaling pathways further downstream of GGPP supplementation. Indeed, GSK3 restriction in MKD patients was able to fully rescue IL-10 production (Fig. 6e). As with our previous data, supplementation with squalene was unable to restore IL-10 expression in MKD patients, supporting evidence that the IL-10 defect is specifically due to poor GGPP production (Supplementary Fig. 8e). To examine the dependency of IL-10 expression on BLIMP1 in MKD

patients, given our previous findings in healthy donor B cells, we analyzed the capacity of MKD patients to upregulate BLIMP1 in response to TLR9 stimulation. In spite of the increased proportions of memory B cells present in MKD patients, we demonstrated a reduced capacity to induce BLIMP1 compared to healthy B cells in 2 of 2 donors (Fig. 6f), providing additional mechanistic data to explain why these patients show reduced B cell derived IL-10.

We then addressed if the defect in IL-10 production from MKD patients was associated with a functional impairment. Indeed, we observed that MKD B cells were unable to suppress IFNγ production by autologous $CD4^+$ T cells, when compared to B cells from healthy controls (Fig. 6g). Moreover, pre-treatment of MKD B cells with exogenous GGPP prior to co-culture, was able to restore regulatory capacity in 2 of 2 donors tested (Supplementary Fig. 8f), suggesting that a GGPP deficiency in MKD patients impairs both IL-10 expression and results in defective suppressive function. Given our B-cell phenotype, we examined whether MKD imposed more a systemic defect on IL-10 producing immune cells we investigated whether T-cell responses from MKD patients were also defective in IL-10 expression. In contrast to B cells, T cells were able to induce effective IL-10 responses, alongside comparable levels of other cytokines, including TNF, IFNγ, IL-2, and IL-17 (Supplementary Fig. 8g).

Finally, to corroborate our results, we interrogated an independent dataset (GSE43553 taken from ref. [34]) that performed global gene expression analysis in PBMCs ex vivo, derived from healthy controls ($n = 20$) and MKD patients ($n = 8$). Gene set enrichment analysis identified defective Ras (KRAS) and PI3K-Akt-mTOR signaling pathways in MKD patients when compared to healthy donors, in line with our own observations (Fig. 6h). Collectively, and consistent with our in vitro findings, these data suggest that dysregulated cholesterol metabolism in vivo, as seen in MKD patients, results in impaired regulatory B-cell responses.

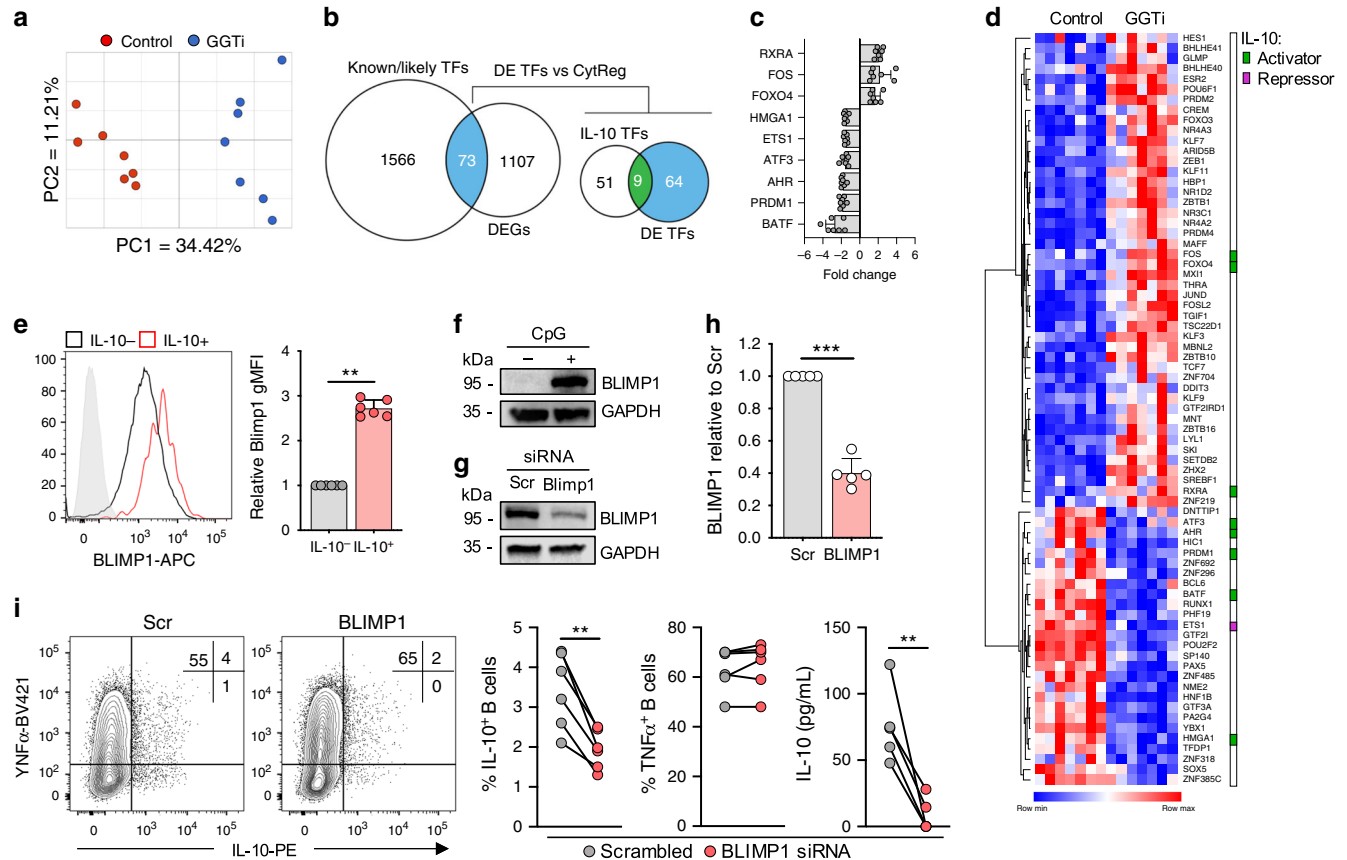

**Fig. 5 GGPP regulates IL-10 induction via BLIMP1. a** Principal component analysis of human B cells after stimulation through TLR9 ± geranylgeranyl transferase inhibition (GGTi), with analysis performed on total normalized counts generated from RNA-sequencing data, before statistical analysis ($n = 7$). **b** Workflow for the identification of putative IL-10 transcription factors. First, differentially expressed genes were cross referenced with known or likely human transcription factors, and the subsequently identified differentially expressed transcription factors were cross referenced with previously validated IL-10 transcription factors in either macrophages or T cells. Differentially regulated IL-10 transcription factors are shown in (**c**) and all differentially regulated transcription factors are shown in (**d**) ($n = 7$). **e** Expression of *BLIMP1* in human B cells within either IL-10$^+$ or IL-10$^-$ B cells after stimulation through TLR9 ($n = 6$, pval = 0.003). **f** Western blot showing BLIMP1 expression in human B cells either in unstimulated or stimulated through TLR9 (CpG) after 40 h (representative of three independent experiments). **g**, **h** Western blot showing expression of BLIMP1 in human B cells after stimulation of TLR9 and nucleofection with either a scrambled control (Scr) or BLIMP1 siRNA ($n = 5$, pval = 0.001). **i** IL-10 and TNF expression in human B cells stimulated through TLR9 after nucleofection with either Scr or BLIMP1 siRNA ($n = 6$ for flow cytometry, pval = 0.002; and 5 for ELISA, pval = 0.009). Each data point represents individual donors. All data presented are mean ± SD where average values are shown. Statistical analysis in all figures was done using a paired *t* test. **$P < 0.01$, ***$P < 0.001$ and all significant values are shown.

## Discussion

This study provides the first description of the metabolic requirements for IL-10 producing B cells, arguing for a central reliance on cholesterol metabolism. Our data point to a model in which synthesis of GGPP prior to stimulation permits transduction of receptor signaling cascades necessary for IL-10 expression. As a consequence, the transcription factor BLIMP1 is induced, which then promotes IL-10 gene expression. We propose that this has direct relevance in vivo, as IL-10 producing B cells from patients who carry mutations in the cholesterol metabolic pathway phenocopy our in vitro findings (outlined in Fig. 7).

Investigations into B-cell metabolism have focused primarily on either antibody production or activation induced metabolic reprogramming[18–20,35]. As alluded to above, there was no understanding of the metabolic requirements for regulatory B cells. Here, we demonstrate that cholesterol metabolism is critical in mediating the regulatory capacity of human B cells through its control of IL-10. We have also previously shown that T cells require an intact cholesterol biosynthesis pathway to switch from inflammatory Th1 cells (IFNγ$^+$IL-10$^-$) to IFNγ$^+$IL-10$^+$ cells[22],

although this appears to be regulated by an alternative mechanism, as inhibition of isoprenylation did not affect this switch. Interestingly, cholesterol metabolism has been implicated in regulatory T-cell function through a mechanism dependent on ICOS and CTLA-4, while having no effect on IL-10 expression[36]. Although the authors did not explore this, we anticipate that regulatory T cells may also possess significant GGPP dependency, as regulatory T-cell function is especially reliant on PI3Kδ activity[37]. Therefore, while cholesterol metabolism appears to regulate different effector molecules between the cell types, it may be that regulatory T and B cells rely more heavily on cholesterol metabolism due to the necessity for potent GGPP-dependent PI3Kδ activity.

Much of the experimental data regarding immunity and metabolism have suggested a paradigm in which the integration of metabolic pathways controlling cell fate arises as a direct consequence of immune cell activation. Based on the ability of cholesterol metabolism to control induction of a regulatory program in human B cells by modulating TLR9 signaling, we propose that the programming of immune responses through cholesterol metabolism may differ in this regard. We propose that isoprenyl modifications

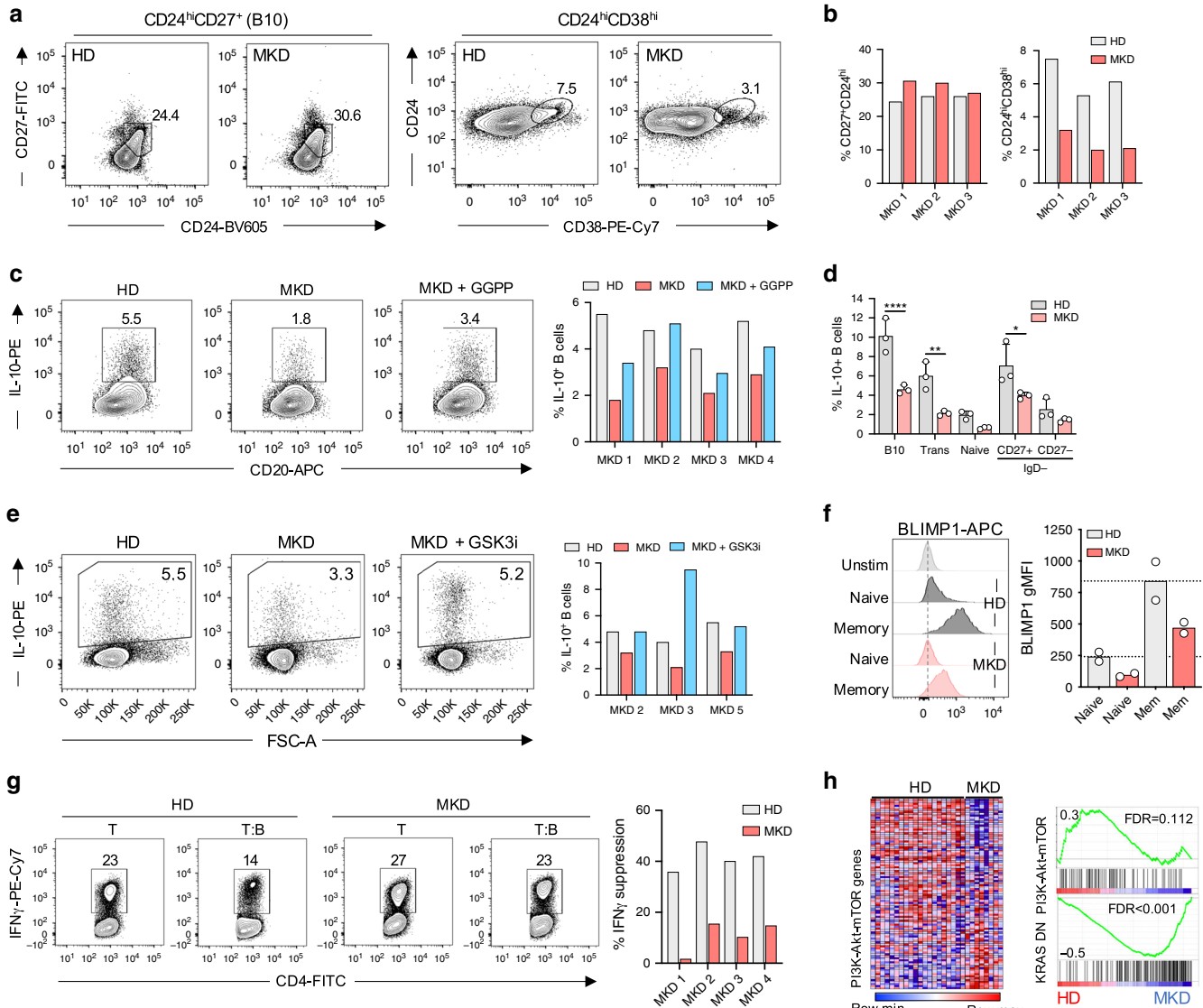

**Fig. 6 MKD patients generate weak regulatory B-cell responses. a**, **b** Frequencies of B10 (CD24hiCD27+) and CD24hiCD38hi regulatory B cells in mevalonate kinase deficient (MKD) patients, compared to age and sex matched healthy donors (HD). **c** IL-10 expression in MKD patients and HD after stimulation through TLR9 ± geranylgeranyl pyrophosphate (GGPP). **d** IL-10 expression within B10, CD24hiCD38hi, naïve, and memory populations, showing the differential contribution of B cell phenotypes to IL-10 production ($n = 3$, pval = <0.0001, 0.002, and 0.01). **e** IL-10 expression in MKD patients and HD after stimulation through TLR9 ± GSK3i (CHIR-99021). **f** BLIMP1 expression in naïve and memory HD or MKD patient B cells after stimulation through TLR9 ($n = 2$). **g** IFNγ production in human CD4+ T cells of healthy controls and MKD patients after co-culture with autologous TLR9-activated B cells. Protocol is outlined in Fig. 1c. **h** Gene set enrichment analysis conducted on gene expression profiles from an independent dataset (GSE43553) on PBMCs ex vivo in MKD patients ($n = 8$) or HD ($n = 20$). Heatmap shows all genes involved in "Hallmark" PI3K-Akt-mTOR gene set collection from MSigDB, and their relative expression in MKD vs. HD. Enrichment plots show PI3K-Akt-mTOR and KRAS DN (genes downregulated by KRAS activation) in MKD vs. HD. Each data point represents individual donors. All data presented are mean ± SD where average values are shown. Statistical analysis in (**d**) was conducted using a paired $t$ test. *$P < 0.05$, **$P < 0.01$, ***$P < 0.001$, ****$P < 0.0001$ and all significant values are shown.

constitute a metabolic preprogramming event. This preprogramming requires the generation of GGPP prior to cellular stimulation (rather than upon or as a direct consequence of stimulation), that has the effect of fine-tuning signaling cascades. This notion of preprogramming is consistent with data showing the GGPP-dependent constitutive localization of Ras family proteins at the cell membrane and endosomes, and that blockade of cholesterol metabolism retains Ras in the cytoplasm[38,39]. In other words, the state of signaling intermediates is preset by the state of cholesterol metabolism of the quiescent cell at any given point.

Our finding that either GGPP deficiency or inhibition of its cognate enzyme GGTase was sufficient to block IL-10 production

suggested that both the metabolite and its enzyme are absolutely required for the induction of a regulatory phenotype. Notably, inhibition of FTase was unable to inhibit IL-10 production, likely due to the differing targets for geranylgeranylation and farnesylation[27]. This suggests that cholesterol metabolism drives the anti-inflammatory function of B cells primarily through the synthesis of the isoprenyl group GGPP. These observations contrast with previous reports suggesting that isoprenylation largely mediates the restriction of pro-inflammatory cytokines, including TNF, IL-6, and IL-1β. It should be noted however, that these data were derived from studies of mouse macrophages and intestinal epithelial cells[23,40,41]. Nonetheless, while human B cells

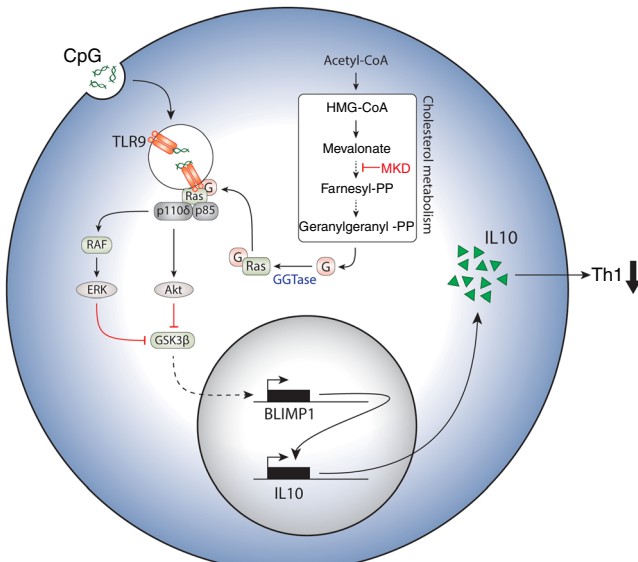

**Fig. 7 B cell IL-10 regulation by GGPP.** IL-10 production by B cells is regulated by cholesterol metabolism though geranylgeranyl pyrophosphate, controlling signaling cascades and BLIMP1 induction downstream of TLR9 engagement. MKD mevalonate kinase deficiency.

are known to co-express inflammatory cytokines TNF and IL-10[42], we saw no alteration in their capability to produce these upon blockade of GGTase. This suggests that pro-inflammatory cytokine production relies less heavily on cholesterol metabolism. Our finding that cholesterol metabolism regulates PI3K signaling may somewhat explain this, given that PI3K can differentially regulate pro- vs. anti-inflammatory cytokine production upon TLR ligation[43–45].

We found that the mechanistic control of cholesterol metabolism revolves around its ability to regulate PI3Kδ-AKT signaling downstream of TLR9, although this is likely not the only target as we also saw an effect on pERK activation. Although we documented expression of all isoforms of PI3K, we observed that PI3Kδ, to a greater extent than PI3Kα or PI3Kγ, contributed to IL-10 expression in human B cells upon TLR9 ligation. This finding is consistent with a recent observation identifying IL-10 producing B cells as the primary pathological cell type in a model of activated PI3Kδ syndrome, driving *Streptococcus pneumoniae* persistence[14]. We further suggest that GSK3 inactivation downstream of GGPP-dependent PI3Kδ-AKT signaling is also required for optimal IL-10 expression. Consistent with this, AKT-driven phosphorylation of GSK3 on Ser9 is known to differentially regulate cytokine production in monocytes[29]. Moreover, we observed that reduced IL-10 expression induced in the context of defective isoprenylation could be rescued through inhibition of GSK3. In line with our findings, GSK3 inhibition in both innate cells[29] and Th1/2/17 cells[46] potentiates IL-10 expression. This provides a common framework in all immune cells, whereby restriction of GSK3 activity upon receptor ligation is necessary for IL-10 expression.

Downstream of TLR9 ligation, BLIMP1 was shown to positively regulate IL-10 expression, suggesting its role as a transcriptional regulator of IL-10 in human B cells. Indeed, BLIMP1 has been previously demonstrated to control IL-10 induction in T cells[47], and in a recent study in regulatory B cells[48]. Analysis of publicly available ChIP-seq datasets in both mouse and human B cells are in line with this finding, suggesting that IL-10 is a significantly enriched target of BLIMP1[33,49]. This observation is reminiscent of recent work highlighting the role for

BLIMP1-expressing regulatory plasmablasts and plasma cells in vivo[3,4,50,51], but also in humans ex vivo[51]. While this previous work provided a correlative link between BLIMP1 and IL-10, we suggest that BLIMP1 is a direct regulator of IL-10 in B cells. Therefore, it will be interesting in future work to further delineate a direct link and understand if there is a common mechanism regulating B cell derived IL-10 in both mouse and human B cells. Surprisingly, we observed expression of BLIMP1 in recently activated IgD+CD27− (naïve) B cells upon TLR9 stimulation (Supplementary Fig. 7F). This has also been observed previously, where expression of BLIMP1 alongside a conformationally open chromatin arrangement in naïve B cells has been demonstrated[52]. This raises the question of whether BLIMP1 plays an earlier role in B-cell function at much earlier stages of activation, distinct from its more classical function linked to the development of terminally differentiated B cells. In addition to BLIMP1, several interesting candidates such as AHR and BATF were also identified in our screen, which warrant further investigation.

To validate our findings in the context of human disease, we investigated patients with dysregulated cholesterol metabolism. MKD patients carry a mutation in the mevalonate kinase gene. As a consequence, their ability to convert mevalonate to mevalonate-5-phosphate is severely impaired. In keeping with our findings following perturbation of cholesterol metabolism in healthy B cells in vitro, we observed poor regulatory responses in B cell from these patients. This anti-inflammatory defect uncovered an unappreciated dimension to the spectrum of MKD. This included a reduced ability to produce IL-10, associated with a functional impairment in restricting T-cell responses. In both cases, supplementation of GGPP was able to reverse this defect, suggesting that a lack of metabolic flux through the pathway might be contributing. Interestingly, GSK3 inhibition was also able to reverse the reduced IL-10 production observed in MKD patients, providing a specific signaling node downstream of GGPP deficiency that is dysregulated in patients. In agreement with GGPP deficiency, MKD patients have been demonstrated to accumulate unprenylated Ras proteins[53]. Further characterization of MKD B cells demonstrated that, despite their increased frequencies of memory B cells, these cells do not effectively induce BLIMP1 expression in response to TLR9. This data is in line with our observations demonstrating that a deficiency in geranylger-anylation results in an inability to properly induce BLIMP1 expression.

To date, the causative factor driving disease pathology has been defined as excessive inflammatory cytokine production, driven through increased macrophage driven IL-1β production[23,54], but also through the induction of a trained immunity phenotype driven by accumulated mevalonate[55]. Here, we also provide evidence that CD4+ T cells in MKD patients induce cytokine responses equivalent to those in healthy donors, suggesting this effect is intrinsic to B cells. Given the role of regulatory B cells in restricting cellular responses, we propose this B cell defect could contribute to the relapsing and remitting nature of the disease. Moreover, it has been shown that B cell-derived IL-10 is important in T follicular helper organization, and restricting excessive antibody responses via interaction with Tfh cells[56,57]. It is therefore tempting to speculate that these patients generate inappropriately inflammatory B-cell responses in lymphoid organs. This would go some way to explain the clinical observations in MKD patients who are typically diagnosed in early childhood with high circulating IgD and IgA levels[25].

In summary, these results argue that cholesterol metabolism acts as a central metabolic pathway promoting an intrinsic anti-inflammatory program in B cells, driving IL-10 production and subsequent suppression and restriction of immune responses.

## Methods

**Cell isolation and culture.** Blood samples were obtained locally from healthy donors, and mevalonate kinase deficient patients were recruited from Royal Free Hospital NHS Foundation Trust. Human: Peripheral blood mononuclear cells were isolated by density gradient centrifugation using Lymphoprep (Alere Technologies). Fresh CD19$^+$ B cells and CD4$^+$ T cells were subsequently isolated by magnetic cell sorting, based on CD19- and CD4-positive or CD4-negative selection (MACS, Miltenyi Biotech); purity was consistently >97%. Cells were cultured in RPMI-1640 (Sigma), 10% Fetal Bovine Serum (Sigma) with 2 mM L-glutamine (Sigma), 0.1 g L$^{-1}$ sodium bicarbonate (Sigma), supplemented with 100 U mL$^{-1}$ penicillin and 0.1 mg mL$^{-1}$ Streptomycin (PAA), and 10 mM Hepes (PAA), in round bottom 96-well Nunclon plates (Thermo Scientific) at $0.3 \times 10^6$ cells/well. B cells were activated with TLR9 ligand CpG ODN 2006 (1 μM, Invivogen), alongside recombinant human IL-2 (25 U ml$^{-1}$, Proleukin, Novartis) typically for 40 h, or as indicated. Where indicated, CellTrace Violet (Thermofisher Scientific) labeling of cells was conducted as per manufacturer's instructions, used at a final concentration of 2 μM prior to stimulation. All inhibitors and metabolites were extensively titrated prior to use, and reagent details are as follows: MK-2206 (Akt, 100 nM), FTi-277 (Farnesyl transferase, 5 μM), GGTi-298 or GGTi-2133 (Geranylgeranyl transferase-II, 5 μM), Psoromic acid (Geranylgeranyl transferase-I, 0.1–10 μM) CHIR-99021 (GSK3α/β, 5 μM), atorvastatin (HMG-CoA reductase, 5 μM), SCH772984 (ERK, 1 μM), HS-173 (p110α, 5 μM), IP-549 (p110α, 1 pM-1 μM), nemiralisib (p110δ, 1 pM–1 μM), (R)-mevalonic acid (250 μM), Squalene (1–10 μM), and geranylgeranyl pyrophosphate (2 μM). Mouse: Total splenocytes were isolated from 8–15-week-old mice, red blood cells were lysed (Biolegend), and resulting single cell suspensions were cultured in the above media in flat bottom 48-well Nunclon plates (ThermoFisher), at $1 \times 10^6$ cells/well. Wild type, p110δ E1020K and p110δ D910A mice used in the study have been described previously[58]. B cells were stimulated with CpG ODN 1826 (1 μM, Invivogen) in the presence of IL-2 (25 U ml$^{-1}$, Proleukin, Novartis) for 48 h. Other ligands used for stimulation include: CD40L (6245-CL, 1 μg ml$^{-1}$, R&D Systems), LPS (10–500 ng ml$^{-1}$, Invivogen), and IFNα (200–1000 U ml$^{-1}$).

**T- and B-cell coculture.** Autologous CD4$^+$ T and CD19$^+$ B cells were isolated from healthy donors or MKD patients as described above. B cells were incubated overnight with CpG (1 μM), IL-2 (25 U ml$^{-1}$), ±atorvastatin, ±mevalonic acid. In parallel, CD4$^+$ T cells were stimulated with plate bound αCD3 and αCD28 (both 1 μg ml$^{-1}$, Biolegend). After 12 h, B cells were washed twice, added to the T cells at a 1:1 ratio, and cultured for 4 days ± αIL10 antibody ± isotype control (5 μg ml$^{-1}$, both Biolegend). Assessment of the direct effect of IL-10 on human CD4 T cells was performed by plate bound αCD3/28 (1 μg ml$^{-1}$) stimulated, cell trace violet labeled T cells, alongside incubation of the indicated concentration of IL-10 for 4 days.

**Flow cytometry.** Antibodies used were as follows: IL-10-PE (dilution 1:800, JES3-19F1), TNF-BV421 (dilution 1:400, Mab11), IFNγ-PE/Cy7 (dilution 1:400, 4S.B3), CD20-AF647 (dilution 1:400, 2H7), CD24-BV605 (dilution 1:400, ML5), CD27-FITC (dilution 1:200, M-T271), CD38-PE/Cy7 (dilution 1:400, HB-7), IgD-BV421 (dilution 1:400, IA6-2), CD4-FITC (dilution 1:200, A161A1), HLA-DR-PerCP/Cy5.5 (dilution 1:200, L243), CD86-BV421 (dilution 1:400, IT2.2), CD40-PE (dilution 1:400, 5C3): all Biolegend, pGSK3ser9-PE (dilution 1:10, REA436, Miltenyi), and BLIMP1-APC (dilution 1:10, 646702, R&D Systems). For intracellular cytokine detection, cells were restimulated in the final 3 (human) or 5 (mice) hours of culture with phorbol 12-myristate 13-acetate (50 ng mL$^{-1}$, Sigma) and ionomycin (500 ng mL$^{-1}$, Sigma) in the presence of Brefeldin A, and GolgiStop (both 1 μl ml$^{-1}$, both BD Biosciences). For surface staining, cells were harvested and incubated with Fixable Viability Dye eFluor780 (dilution 1:4000, eBiosciences) for 15 min in phosphate-buffered saline (PBS), followed by the appropriate volume of antibody diluted in 0.5% bovine serum albumin (BSA) in PBS for 20 min, all at 4 °C. Cells were then washed and fixed in 3% paraformaldehyde (Electron Microscopy Sciences) for 15 min at room temperature. For intracellular cytokine staining, cells were incubated with the appropriate volume of antibody, diluted in 0.1% saponin in 0.5% BSA in PBS for 45 min at room temperature or 4 °C overnight. For staining of transcription factors, or phospho-flow, FoxP3/Transcription factor buffer set was used, as per manufacturer's instructions (eBiosciences). Staining using a secondary antibody against the primary was undertaken either at room temperature for 1 h, or 4 °C overnight. Cells were acquired using a BD LSRFortessa or FACSCanto II (BD Biosciences), and analysis conducted using FlowJo V.10.1 software (Tree Star Inc.).

**RNA extraction and qRT-PCR analysis.** Harvested cells were lysed in TRIzol (Ambion, Life Technologies), and stored at −20 °C until RNA extraction by phenol–chloroform phase separation. Briefly, RNA was eluted with chloroform, and subsequently precipitated using ice cold isopropanol for 1 h at −20 °C. If necessary, RNA clean-up was conducted by repreciptation of RNA in 0.5 volumes of 7.5 M ammonium acetate, 2.5 volumes 100% ice-cold ethanol, and 1 μl GlycoBlue overnight at −20 °C. Totally, 40 ng of RNA was added for reverse transcription, using qPCRBIO cDNA Synthesis Kit (PCR Biosystems) as per manufacturer's instructions. qPCR was conducted using 7900HT Fast Real-Time PCR System

(ThermoFisher Scientific). Primers used were: IL10 (Hs00961622_m1), IL6 (Hs00174131_m1), or LTA (Hs04188773_g1), multiplexed with a VIC- labeled 18S probe (all Thermofisher Scientific) in PCR Biosystems mastermix.

**ELISA.** Sandwich ELISA was used to detect supernatant analytes. All cell free supernatants harvested at indicated time points were stored at −20 °C until analysis. ELISAs for IL-10, IFNγ (both R&D systems, cat nos. DY217B and DY285B, respectively), TNF (Biolegend, cat no. 430201), or IgM (ThermoFisher Scientific, cat no. BMS2098) were conducted according to manufacturer's protocols and detected on a Victor 1420 multilabel counter (Perkin Elmer) quantifying concentrations drawn from a standard curve on each plate.

**Western Blotting.** Antibodies used were as follows: BLIMP1 (dilution 1:1000, 646702, R&D Systems), pAkter473 (dilution 1:1000, D9E), pERKThr202/Tyr204 (dilution 1:1000, D12.14.4E), and GAPDH (dilution 1:5000, 14C10, all Cell Signaling Technologies). After culture, cells were harvested, washed in ice-cold PBS, and lysates immediately extracted, or pellets were snap frozen on dry ice for 1 min and kept at −80 °C until lysis. Whole cell lysates were extracted using RIPA Buffer (Cell Signaling Technology) with 1× Protease inhibitor cocktail. Lysates were added to 4× Laemmli Sample Buffer (Bio Rad) with 10% 2-mercaptoethanol, and resolved on sodium dodecyl sulfate polyacrylamide gel electrophoresis gels, transferred to polyvinylidene fluoride membranes, blocked (Tris, 5% BSA, 0.05% Tween20) and probed with the desired antibody. Immunoblots were developed with anti-rabbit-HRP (Dako) secondary antibody, and proteins visualized by SuperSignal chemiluminescent reaction (Pierce) in a ChemiDoc station (BioRad).

**siRNA knockdown.** siRNA knockdown was conducted by Amaxa nucleofection using the nucleofector 2b machine, as per manufacturers protocol. siRNA knockdown was optimized, resulting in the used of 2–3 million cells per condition, and 500 nM of either BLIMP1 targeted siRNA (Silencer Select assay ID s1992) or scrambled control (Silencer Select Negative control, both from ThermoFisher Sceintific). Cells were electroporated using program U-015, resuspended in warm fully supplemented culture medium, and left to recover for 15 min before stimulation. Knockdown efficiency was determined by western blot at 40 h post stimulation.

**RNA sequencing.** CD19$^+$ B cells were stimulated with CpG (1 μM) and IL-2 (25 U ml$^{-1}$) ± GGTi-298 for 12 h. RNA was isolated by column centrifugation using RNeasy Plus mini kit (with gDNA removal step) as per manufacturer's instructions (Qiagen). Library preparation was completed using NEBNext Ultra Directional RNA Library Prep Kit for Illumina. Depletion of ribosomal RNA was performed using Next rRNA Depletion kit (New England BioLabs). RNA quality was confirmed by bioanalyser (Agilent 2100 Bioanalyzer G2938B), resulting in a mean RIN score of 8.1, ranging from 7.3 to 8.7. Paired-end sequencing was then conducted using the HiSeq 2500 platform (Illumina). Raw data were checked for quality using FASTQC. Processing of the raw data involving alignment (STAR) and annotation (hg19) were done using Partek. After annotation, TMM-normalized data were used for downstream statistical analysis using the exact test in the edgeR package. Inclusion criteria for significantly differentially expressed genes was a false discovery rate of <0.05 and a fold change of greater than 1.5×. Subsequent processing and visualization of the data was completed in RStudio or Morpheus (Broad Institute, Boston, MA).

**Gene set enrichment analysis.** Gene set enrichment analysis used on dataset GSE43553[34] was conducted with GSEA v4.0.2 (Broad Institute). Gene expression data was used from 20 healthy controls (GSM1065214 to GSM1065233) and 8 MKD patients (GSM1065234 to GSM1065241). Gene sets were compared to the gene set database h.all.v7.0.symbols.gmt [Hallmarks], using 1000 permutations, the "weighted" enrichment statistic, and "Signal2Noise" for gene ranking.

**Statistics.** All statistical analysis was conducted using GraphPad Prism v7.0 (GraphPad, San Diego, CA, USA). Paired data were analyzed through either two-way ANOVA with Dunnett's test for multiple comparisons, or Freidman's test with Dunn's test for multiple comparisons. Unpaired data were assessed through ANOVA with Tukey's test for multiple comparisons. Finally, two-group comparison was performed either through unpaired or paired $t$ test. A $P$ value of <0.05 was considered statistically significant, and all statistically significant values are shown.

**Study approval.** Blood samples were obtained locally from healthy donors, and mevalonate kinase deficient patients were recruited from Royal Free Hospital NHS Foundation Trust, London, UK or National Human Genome Research Institute/ Inflammatory Disease Section, Bethesda, USA. Written informed consent was received from all healthy donors and patients used in this study, approved by the Bromley Research Ethics Committee (REC06/Q0705/20) and by the NIDDK/ NIAMS Institutional Review Board. Animal experiments were performed according to the Animals (Scientific Procedures) Act 1986, license PPL 70/7661 and approved by the Babraham Institute Animal Welfare and Ethics Review Body.

**Reporting summary**. Further information on research design is available in the Nature Research Reporting Summary linked to this article.

## Data availability

Sequence data generated in this study have been deposited in GEO under the accession code GSE150245. Publicly available datasets used for analysis are available at GEO under the accession codes GSE43553 (used in GSEA analysis), and GSE71698 (used for ChIP and RNA-seq analysis). Source data are provided with this paper.

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

## Acknowledgements

We thank all the healthy donors and patients for their support. This research was funded/supported by the National Institute for Health Research (NIHR) Biomedical Research Centre based at Guy's and St. Thomas' NHS Foundation Trust and King's College London and/or the NIHR Clinical Research Facility. The views expressed are those of the author(s) and not necessarily those of the NHS, the NIHR or the Department of Health and Social Care This work was also supported by: the IMI-funded project BeTheCure (115142-2); EU/EFPIA Innovative Medicines Initiative 2 Joint Undertaking RTCure (777357) (both E.P. and A.P.C.); the National Institute for Health Research Biomedical Research Centre, at Guy's and St. Thomas' NHS Foundation Trust and King's College London (J.A.B.); Intramural Research Programs of the National Human Genome Research Institute (I.A. and S.R.); MRC (MR/M012328/2, K.O.); Wellcome Trust (206618/Z/17/Z, A.C.)

## Author contributions

J.A.B. designed, performed, and interpreted the experiments, and wrote the paper, H.A.P. contributed to experimental design, performed experiments, and contributed to the paper preparation, T.H. performed the experiments, A.C. and K.O. contributed to experimental design and provided the mice, M.W., S.R., H.J.L., and I.A. recruited the patients and extracted patient the samples, H.J.L. contributed to the experimental design and provided the patient samples, C.K. supervised research, and contributed to the experimental design, A.P.C. and E.P. conceived the project, designed and performed experiments, supervised research, and contributed to the paper preparation.

## Competing interests

The authors declare no competing interests.
