## [Peer Review File · Nature Communications]

Reviewers' comments:

Reviewer #1 (Remarks to the Author):

In this manuscript, Bibby et al propose that the induction of a regulatory program in human B cells is dependent on cholesterol metabolism. They show that synthesis of the metabolic intermediate geranylgeranyl pyrophosphate (GGPP) was required to specifically drive IL-10 production. Furthermore, GGPP-dependent protein modifications controlled signaling through PI3K, which in turn promoted BLIMP1-dependent IL-10 production.

This is fundamentally an interesting topic and this reviewer is prepared to accept that the pathways proposed by the authors impact IL-10 production from B lineage cells. However, in its current form the study falls short of providing clarity but rather confuses the issue. Both IL-10 and Blimp1 expression are tightly linked to discrete differentiation stages of B lineage cells (ie plasma blasts and plasma cells). Thus, it is not correct to explore these pathways without taking this into account.

Major points

IL-10 production from B lineage cells has been shown to be largely restricted to plasma blasts (eg Matsumoto et al Immunity 2014). Similarly, Blimp1 expression is limited to plasma cells as widely documented. Firstly, this needs to be acknowledged and discussed in the manuscript. The authors state that "Currently there is no defined transcription factor that regulates IL-10 in human B cells" While this maybe true for human B cells, the role of Blimp1 in IL-10 production from many different cell types is well established for mouse.

Secondly, it is essential to examine how the proposed pathways (inhibition of HMG-CoA reductase or GGTase) impact on plasma cell differentiation and other basic processes such as proliferation and survival. Antibody production should be part of the parameters measured. The authors look at proliferation and survival, but only after 40h. This is too early to expect to see major differences. What happens at later time points?

Overall, I am concerned about the main readout in this manuscript. In most experiments the population of IL10+ B cells is very small (2-3%) and a substantial fraction of this small population sits at the very low end of the gate and may actually constitute background (eg Fig. 3G or Fig 5I). This makes me wonder how relevant the population is in vivo and whether or not the authors measure the right population/time point?

In this context: Fig. S4A shows a very distinct population of IL-10+ B lineage cells that sits separate from all other cells. What are these cells? The interpretation of the authors that 'all B cell populations contribute to IL-10 production' is hard to follow given this result. If the authors were able to pinpoint this population and show how it is specifically regulated, this would make the study much more attractive. It just does not make sense to claim that naïve and transitional B cells produce IL-10 in a Blimp1 dependent manner. These cells do not express Blimp1.

Finally, MKD patient B cells need to be fully characterized with respect to the above listed parameters and pathways. It is clear that these patients have a completely altered B cell compartment, both in terms of subset composition and activation status. Hence, with clear dissection of the role of differentiation, it is impossible to draw firm conclusions.

Reviewer #2 (Remarks to the Author):

This study provides evidence that geranylgeranylation is important for TLR9-mediated induction of

IL10-producing human B cells. This is shown using inhibitors of HMGCoA reductase, GGTase and FTase, and analysis by FACS, qPCR and ELISA. Induction of another cytokine, TNF, is unaffected by the inhibitor treatments. TLR9-mediated activation of Akt and to a lesser extent Erk is antagonized by the GGTi and GSK is identified as a likely enzyme that needs to be repressed by Akt during IL10 induction. Inhibitor and genetic experiments show that PI3Kd is required for TLR9-induced IL10. Gene expression analysis of TLR9 activated control and GGTi treated B cells leads to identification of several TFs as being induced in a geranylgeranyl-dependent manner. Blimp1 is focused on as IL10+ B cells have previously been shown to express Blimp1, and shRNA mediated knockdown of Blimp1 leads to reduced IL10 but not TNF. Finally, B cells from human patients with hypomorphic mutations in mevalonate kinase (MKD patients) are shown to make less IL10 upon CpG stimulation. Overall, this study contains a set of interesting insights about requirements for TLR9-induced IL10 production in B cells. However, there are also a number of weaknesses that currently limit enthusiasm. Since the study is focused on human B cells, it is performed entirely in vitro, which is understandable. However, the extensive reliance on chemical inhibitors raises concerns about off target effects. More generally, while geranylgeranylation is implicated as being important in regulatory B cell IL10 production, the key question of what protein(s) needs to be geranylgeranylated is not answered.

Specific comments:

1. The GGTi used in this study (GGTi-298) inhibits geranylgeranyl transferase I. The authors didn't test the inhibition of GGTaseII (Rab geranylgeranyl transferase). Although GGTaseII inhibitors are less available, it is notable that the authors used one such inhibitor, psoromic acid, in a study earlier this year (Nat Commun 10, 498). Moreover, given the concerns about inhibitor off-target effects, it seems important to show that key findings made with single inhibitors can be confirmed where possible with a second inhibitor for that enzyme (or by a second approach as in 2).
2. The data in Figure 5F show the ability of the authors to achieve knockdown of gene expression in human B cells. Given the central conclusion of the study regarding the role of GGTase and geranylgeranylation in TLR9-mediated IL10 induction, it would seem appropriate to show that this result is obtained in siRNA-mediated knockdown cells (and not just with chemical inhibitors).
3. Since the mevalonate pathway has two main outcomes (cholesterol and isoprenoids), it would seem appropriate to demonstrate whether or not addition of a cholesterol-committed metabolite such as squalene to the statin treated cells has any effect on IL-10.
4. Given the crucial role of small G-proteins in many B cell activation events, one wonders if the selective effect observed in this study is the dependence of IL10 on such isoprenylated proteins or, rather, the lack of TNF dependence on them. The effect of GGTi treatment on various other B cell activation outputs (other cytokines, activation marker expression, proliferation, viability) needs to be shown.
5. The authors state: "Isoprenyl modifications almost exclusively regulate the localization of Ras superfamily proteins", and so investigated selected Ras family signaling pathways downstream of TLR9. However, the cited review does not make this statement and the quantitatively dominant targets of geranylgeranylation are still poorly defined.
6. It is stated: "Gene set enrichment analysis identified defective Ras (KRAS) and PI3K-Akt-mTOR signaling pathways in MKD patients when compared to healthy donors, in line with our own observations (Figure 6F)". However, KRAS is farnesylated not geranylgeranylated according to the cited review (it can be geranylgeranylated, but usually when FTase is first inhibited or GGTaseI is overexpressed (cross-prenylation)). Also, when interrogating these data, the ERK signaling shown earlier to have an effect on IL-10 is not mentioned.
7. It is stated that addition of GGPP rescues the defective IL10 production in CpG stimulated MKD B cells. However, the rescue does not appear to be statistically significant (Fig. 6C). While it is recognized that the patients are rare, it would have been more compelling if additional data (such as technical repeats) were available. Comparison to the effect of squalene addition would also have been informative.

Minor

1. "Cholesterol metabolism" as referenced in the title seems to vague given that the paper focuses

on GGPP (and protein prenylation) rather than cholesterol. A more appropriate title might be:
"GGPP is required for IL-10 production by regulatory B-cells"

2. The name of the FTi inhibitor and the ERK inhibitor needs to be provided. The names of the inhibitors should be provided in the main text or figure legends and not just in the methods.

3. The focus on PI3K-AKT as one of the downstream pathways is reasonable but this is quite possibly just one of many downstream pathways involved. It is important not to suggest that there is a selective role of geranylgeranylation in the activation of this pathway. For example, the data show that there is a reduction in pERK as well as pAKT.

4. The differences between these findings for B cell IL10 expression and the earlier findings of cholesterol biosynthesis requirements in T cell IL10 production (Nat Commun 10, 498) is not adequately discussed.

Reviewers' comments:

Reviewer #1 (Remarks to the Author):

In this manuscript, Bibby et al propose that the induction of a regulatory program in human B cells is dependent on cholesterol metabolism. They show that synthesis of the metabolic intermediate geranylgeranyl pyrophosphate (GGPP) was required to specifically drive IL-10 production. Furthermore, GGPP-dependent protein modifications controlled signaling through PI3K, which in turn promoted BLIMP1-dependent IL-10 production.

This is fundamentally an interesting topic and this reviewer is prepared to accept that the pathways proposed by the authors impact IL-10 production from B lineage cells. However, in its current form the study falls short of providing clarity but rather confuses the issue. Both IL-10 and Blimp1 expression are tightly linked to discrete differentiation stages of B lineage cells (ie plasma blasts and plasma cells). Thus, it is not correct to explore these pathways without taking this into account.

We thank the reviewer for their positive feedback and helpful comments. The reviewer raises important points that broadly focus on the differentiation of B cells, and whether cholesterol metabolism affects this process, in addition to its effect on IL-10. Furthermore, the reviewer raises the question as to whether these processes are also dysregulated in patients with mevalonate kinase deficiency. We agree that BLIMP1 is tightly linked to B cell differentiation, but the evidence for lineage specific expression of IL-10 is less well defined. This is especially true in humans, where we and others show that IL-10 can be expressed during all stages of the B cell life cycle from immature B cells to fully differentiated plasma cells (Supplementary Figure 4a-c, and Figure 6d, and this diverse literature is well outlined in Rosser and Mauri, *Immunity*, 2015).

We initially focussed on phenotyping B cells in shorter term cultures (Supplementary Figure 4d-f) due to the kinetics of IL-10 expression that we see after stimulation, which peak at around 36 hours (Figure 1a). In response to the reviewer's suggestion, we addressed the contribution of cholesterol metabolism and geranylgeranylation in longer term cultures (5-7 days) in both healthy donors and mevalonate kinase deficient patients.

With regards to specific B cell populations being regulated by isoprenylation, we have shown in both healthy donors and MKD patients that the geranylgeranylation dependent signalling pathway driving IL-10 expression is common across all B cell populations measured, and is not restricted to a specific subset. We see that GGase activity is required for IL-10 expression in naïve, memory, CD24^{hi}CD38^{hi}, and B10 populations. This highlights a particularly interesting feature; notably, that all of these populations utilise a common mechanism to induce a regulatory phenotype. This point is outlined in the manuscript under the '*Cholesterol metabolism drives IL10 independent of B cell population*' results subheading and is also observed in MKD patients in Figure 6d.

A more detailed, point-by-point response to the reviewer's comments is provided below, and all changes are also highlighted in green text within the manuscript file.

Major points

IL-10 production from B lineage cells has been shown to be largely restricted to plasma blasts (eg Matsumoto et al Immunity 2014). Similarly, Blimp1 expression is limited to plasma cells as widely documented. Firstly, this needs to be acknowledged and discussed in the manuscript. The authors state that "Currently there is no defined transcription factor that regulates IL-10 in human B cells" While this maybe true for human B cells, the role of Blimp1 in IL-10 production from many different cell types is well established for mouse.

We thank the reviewer for highlighting what has become an important issue in the literature. Namely, that there are differences between murine and human data. Whilst work assessing murine IL-10 expressing B cells has indeed focussed on plasmablasts (we have included discussion around this point on lines 340-346), the relative contribution of IL-10 production by naïve and memory B cells in humans has shown varied results. For example, the majority of studies addressing human IL-10 expressing B cells has focussed on CD24^{hi}CD38^{hi} immature B cells (e.g. see Menon, M, Immunity 2016, Blair, PA, Immunity 2010, and a review highlighting clinical data around regulatory B cells in Mauri and Menon, JCI, 2017).

Regarding BLIMP1 control of IL-10, we have incorporated the reviewer's comment that BLIMP1 has previously been shown to regulate IL-10 in other cell types by including additional text in the manuscript on lines 337-339.

Whilst we acknowledge the large amount of literature assessing BLIMP1 expression in plasma cells, we are unaware of any data analysing the expression of BLIMP1 in human B cell populations after TLR9 ligation. There has however been some suggestion that naïve human B cells can express BLIMP1 (see Jenks, SA, Immunity, 2018). We therefore analysed the effect on BLIMP1 expression after stimulation with CpG. Surprisingly, and albeit at much lower levels than in memory populations, we saw clear upregulation of BLIMP1 in naïve and both B10 and CD24^{hi}CD38^{hi} B cell populations (we have included this data in Supplementary Figure 7f, and added a description and discussion of this data on lines 204-206 and 346-351 including a reference to the Jenks, SA et al. publication). We hypothesise that this low expression of BLIMP1 in naïve cells may be due to one of several reasons: that BLIMP1 expression is bimodal after TLR9 activation, or that those cells expressing BLIMP1 at low levels are transitioning and differentiating into memory B cells. However, as our data seem to suggest that B cell differentiation does not appear to be a central determinant of IL-10 expression in human B cells, and since we demonstrate that geranylgeranylation dependent IL-10 production is common to all B cell populations, we have not investigated this further. Nonetheless, upregulation of BLIMP1 across B cell populations supports our data that IL-10 is driven by BLIMP1 upon TLR9 activation.

Secondly, it is essential to examine how the proposed pathways (inhibition of HMG-CoA reductase or GGTase) impact on plasma cell differentiation and other basic processes such as proliferation and survival. Antibody production should be part of the parameters measured. The authors look at proliferation and survival, but only after 40h. This is too early to expect to see major differences. What happens at later time points?

To answer this point, we have analysed the contribution of cholesterol metabolism generally, and also geranylgeranylation specifically, to the above processes. Specifically, we cultured cells for 5-7 days after TLR9 ligation in the presence of either atorvastatin or GGTi, and subsequently analysed proliferation, differentiation, viability (all 5 days), and antibody production (7 days). We found that incubation of cells with atorvastatin had no effect on either proliferation, differentiation, viability, or antibody production. Similarly, GGTi had no effect on viability or antibody production, and minimal effects on proliferation and differentiation. We therefore conclude that this isoprenylation event selectively drives the induction of a regulatory phenotype in B cells that is independent of their differentiation. We have added this data to Supplementary Figure 5c, and have added text relating to this on lines 135-137.

Overall, I am concerned about the main readout in this manuscript. In most experiments the population of IL10+ B cells is very small (2-3%) and a substantial fraction of this small population sits at the very low end of the gate and may actually constitute background (eg Figure 3G or Figure 5I). This makes me wonder how relevant the population is in vivo and whether or not the authors measure the right population/time point?

We acknowledge that the proportion of B cells expressing IL-10 is relatively small. However, flow cytometry plots in the manuscript are accompanied by matched cell culture supernatant ELISA readouts that show significant and physiologically relevant levels of IL-10 being secreted by these cells (between 200-1500ng/ml of IL-10; Figure 1b, and data mostly provided in supplementary figures e.g. Supplementary Figure 3a for atorvastatin treatment or Supplementary Figure 5a for GGTi treatment). As shown in our co-cultures, these levels are able to affect Th1 induction in CD4 T cells (Figure 1d). These expression levels are also in agreement with literature on human B cells, demonstrating similar proportions of IL-10+ B cells, and secreted levels of IL-10 after stimulation (e.g. Blair, PA *et al.*, *Immunity*, 2010 or Iwata, Y *et al.*, *Blood*, 2011). To address the reviewer's comments on specific data in the manuscript, we have included ELISA data corresponding to Figure 3g in Supplementary Figure 6c and referenced in the text on line 166. Regarding Figure 5i, we have provided secreted IL-10 levels measured by ELISA (third panel within Figure 5i) data together with flow cytometry percentages of IL-10+ B cells.

With regards to time point, we see maximal IL-10 expression at around 36 hours (Figure 1a), after which gene expression decreases. We therefore feel confident that the timepoint of 40 hours is appropriate to see TLR9-induced IL-10 expression in B cells.

In this context: Fig. S4A shows a very distinct population of IL-10+ B lineage cells that

sits separate from all other cells. What are these cells? The interpretation of the authors that ‘all B cell populations contribute to IL-10 production’ is hard to follow given this result. If the authors were able to pinpoint this population and show how it is specifically regulated, this would make the study much more attractive. It just does not make sense to claim that naïve and transitional B cells produce IL-10 in a Blimp1 dependent manner. These cells do not express Blimp1.

We thank the reviewer for highlighting this. We clarify Supplementary Figure 4a as follows. The tSNE clustering algorithm has separated this distinct population of B cells precisely because IL-10 is the major distinguishing feature (IL-10+ cells are overlaid on the tSNE plot of the bulk B cell population, by highlighting them in red in the first panel). When looking at the cell surface phenotype of this distinct cluster, we see that it is a heterogeneous population consisting of B10, CD24^{hi}CD38^{hi}, naïve, memory B cells, and plasmablasts (each of these population gates are overlaid on the same data and highlighted in red in the subsequent panels 2-6). Therefore, in panels 2-6 we can see that all of these populations appear within the distinct population at the bottom of the tSNE plot characterised by IL-10 expression (graphical representation shown in RevFig 1). This is in agreement with the data in Supplementary Figure 4c, showing that all B cell populations contribute to the pool of cells expressing IL-10. We have amended the figure legend of Supplementary Figure 4a to make this more explicit, and hope that this clarifies the message we are trying to convey.

Regarding BLIMP1, we see little expression *ex vivo* (Figure 5f and Supplementary Figure 7e), although stimulation through TLR9 induces BLIMP1 expression in all

RevFig 1. tSNE analysis, showing gated red populations. Total B cells are represented as grey populations in each tSNE plot. Specific populations within the total B cell population are then overlaid and shown in red, based on the gating shown above each plot. All data shown here are from a single donor

populations of B cells that we analysed. As mentioned above, this includes naïve, memory, B10, and CD24^{hi}CD38^{hi} B cells. In agreement with the literature, we see the lowest expression of BLIMP1 in naïve B cell, and the highest expression of BLIMP1 in memory B cells. Both B10 and CD24^{hi}CD38^{hi} B cells show intermediate expression of BLIMP1 when compared to naïve and memory populations upon TLR9 ligation. We have now added this data to Supplementary Figure 7e-f and included corresponding text on lines 204-205, and 346-351. We believe that the data support the notion of BLIMP1 dependency in regulating IL-10 expression across different B cell populations.

Finally, MKD patient B cells need to be fully characterized with respect to the above listed parameters and pathways. It is clear that these patients have a completely altered B cell compartment, both in terms of subset composition and activation status. Hence, with clear dissection of the role of differentiation, it is impossible to draw firm conclusions.

We have addressed the reviewer's comments by studying B cell activation, differentiation, and subset composition in MKD patients. Again, we analysed longer term cultures to look at proliferation, differentiation, viability, and antibody production in MKD B cells. Broadly, we see the same proliferation capacity, differentiation, and antibody production when compared to healthy donors. Similarly, B cells from MKD patients show equivalent viability after stimulation (all these data are shown in Supplementary Figure 8c, and corresponding text is added on lines 222-224). We therefore conclude that B cells from MKD patients show a relatively normal phenotype at a global level but differ in their capacity to induce a regulatory program after TLR9 ligation.

At the reviewer's suggestion, we also analysed the capacity for MKD patients to upregulate BLIMP1 upon TLR9 ligation. Surprisingly, despite having slightly increased memory B cell frequencies (Supplementary Figure 8b-c), we saw a reduced capacity to upregulate BLIMP1 in response to TLR9 stimulation in both naïve and memory MKD B cell populations (this data is now included in Figure 6f, and corresponding text can be found on lines 242-247). We hypothesise that this may be a contributing factor in the reduced capacity to express IL-10. We thank the reviewer for suggesting this experiment.

To further address the above pathways, we analysed the role of GSK3 in MKD patients. Our data in healthy donors suggested that Akt mediated inhibition of GSK3 was important for IL-10 expression. To further link this signalling pathway to MKD patients, we therefore asked if, like GGPP, GSK3 inhibition was able to rescue IL-10 expression. Indeed, inhibition of GSK3 in MKD B cells was able to rescue IL-10 expression, a finding in keeping with our data in healthy donors showing that GSK3 blockade can rescue IL-10 deficiency induced upon GGTi inhibition (this data is now included in Figure 6e and referenced in the text on lines 236-239 and lines 363-365). We believe this strengthens our data on GGPP supplementation of MKD B cells, linking this with a defect in downstream signalling mediated through GSK3.

As the patients are extremely rare, we have focussed our experiments on the above pathways and processes mentioned by the reviewer and highlighted in our previous findings. Given the strong similarities between healthy donor B cells defective in GGTase activity and B cell from MKD patients, we hope that this additional data provides a convincing *in vivo* correlate of the findings generated *in vitro* using B cells from healthy donors.

Reviewer #2 (Remarks to the Author):

This study provides evidence that geranylgeranylation is important for TLR9-mediated induction of IL10-producing human B cells. This is shown using inhibitors of HMGCoA reductase, GGTase and FTase, and analysis by FACS, qPCR and ELISA. Induction of another cytokine, TNF, is unaffected by the inhibitor treatments. TLR9-mediated activation of Akt and to a lesser extent Erk is antagonized by the GGTi and GSK is identified as a likely enzyme that needs to be repressed by Akt during IL10 induction. Inhibitor and genetic experiments show that PI3Kd is required for TLR9-induced IL10. Gene expression analysis of TLR9 activated control and GGTi treated B cells leads to identification of several TFs as being induced in a geranylgeranyl-dependent manner. Blimp1 is focused on as IL10+ B cells have previously been shown to express Blimp1, and shRNA mediated knockdown of Blimp1 leads to reduced IL10 but not TNF. Finally, B cells from human patients with hypomorphic mutations in mevalonate kinase (MKD patients) are shown to make less IL10 upon CpG stimulation. Overall, this study contains a set of interesting insights about requirements for TLR9-induced IL10 production in B cells. However, there are also a number of weaknesses that currently limit enthusiasm. Since the study is focused on human B cells, it is performed entirely *in vitro*, which is understandable. However, the extensive reliance on chemical inhibitors raises concerns about off target effects. More generally, while geranylgeranylation is implicated as being important in regulatory B cell IL10 production, the key question of what protein(s) needs to be geranylgeranylated is not answered.

We thank the reviewer for their interest in our manuscript, and the helpful suggestions provided. As the reviewer highlights, most of the work is conducted using human B cells either from healthy donors or MKD patients. As we were cautious with regards to off target effects with small molecule inhibitors, we undertook extensive titration of each inhibitor that we used; an example of which is seen in Figure 4b. Wherever possible, we also undertook downstream rescue experiments to reduce the likelihood that our phenotype was due to off target effects. For example, mevalonate supplementation alongside atorvastatin treatment rescues the metabolic pathway directly below the inhibition, and was able to rescue our phenotype. We then showed that supplementation of GGPP – rescuing specifically the geranylgeranylation branch – was also able to reverse the inhibition, in agreement with there being no off-target effects of either atorvastatin or the GGTase inhibitor used here. Furthermore, not being able to rescue our IL-10 phenotype with GGPP supplementation after GGTase inhibition demonstrates that GGPP is specifically being utilised by GGTase (Figure 2e).

We have also conducted similar experiments demonstrating that GSK3 inhibition can rescue inhibition of GGTase and Akt, both of which are upstream of GSK3. MKD patients, who have inherited defects in cholesterol metabolism, provided us with a system that did not rely on chemical inhibition. Furthermore, these patients have previously been shown to have a general defect in isoprenylation of Ras superfamily proteins due to reduced metabolic flux through the pathway (Jurczyk et al. Immunol Cell Biol, 2016 and Munoz et al. JACI, 2017). The ability to rescue IL-10 expression in MKD patients through supplementation of GGPP or GSK3 inhibition lends strong support to our data using healthy donor B cells, where chemical inhibitors were used.

We agree that siRNA mediated knockdown of GGTase would provide alternative supporting evidence. Unfortunately, in spite of multiple attempts, for technical reasons this was not possible. Instead, and as suggested by the reviewer, we have used a different inhibitor of GGTase that showed the same results. As outlined in the point by point responses below, we feel that the additional experiments regarding pathway metabolites conducted as suggested by the reviewer have strengthened our data, and provide additional insights into the specificity of GGTase control of IL-10 in human B cells.

Like the reviewer, we have also been interested in identifying the specific prenylated protein that is involved in this pathway downstream of TLR9. As a starting point, we documented expression of 74 Ras proteins in resting B cells (see RevFig 2 below), a large number of which could be regulated by GGPP. In light of these data, we did not feel that undertaking a systematic study of each Ras family protein was realistic within a short timeframe. We also felt that since our investigations were driven by the initial finding that upstream metabolic pathways were responsible for regulating IL-10, identifying a specific Ras family member or members would not alter the conclusions or impact of our findings.

Below we address in more detail each of the reviewer's suggestions and outline these experiments in a point by point reply. All changes are also highlighted in green text

RevFig 2. Ras superfamily protein expression in resting B cells. Data shown are normalized expression values of Ras superfamily proteins, highlighted by their subfamily in resting B cells.

within the manuscript file.

Specific comments:

1. The GGTi used in this study (GGTi-298) inhibits geranylgeranyl transferase I. The authors didn't test the inhibition of GGTaseII (Rab geranylgeranyl transferase). Although GGTaseII inhibitors are less available, it is notable that the authors used one such inhibitor, psoromic acid, in a study earlier this year (Nat Commun 10, 498). Moreover, given the concerns about inhibitor off-target effects, it seems important to show that key findings made with single inhibitors can be confirmed where possible with a second inhibitor for that enzyme (or by a second approach as in 2).

We thank the reviewer for highlighting the absence of GGTaseII data, and have therefore conducted this experiment using psoromic acid. We see no effect of psoromic acid on IL-10, suggesting that IL-10 expression is dependent on GGTaseI but not GGTaseII activity. These data have now been added to the manuscript in Supplementary Figure 5d and referenced in the text 137-139.

We share the reviewers concerns about using chemical inhibitors and have therefore gone to some length to design experiments in a way that would mitigate against drawing imprecise conclusions. As mentioned above, a good example are the rescue experiments that we utilised: both mevalonate and GGPP, but not FPP, are able to rescue atorvastatin treatment (Figure 1e and Figure 2d); GGPP is unable to rescue GGTi treatment (Figure 2e); GSK3i is able to rescue GGTi treatment (Figure 3g).

In light of the reviewer's suggestions, we have conducted additional experiments using a different GGTi inhibitor (data added in Supplementary Figure 5b, and referenced in the text on lines 131-133), and have also attempted to rescue the atorvastatin phenotype using the cholesterol-committed metabolite squalene (data added in Supplementary Figure 5f and referenced in the text on lines 143-145). These data are in keeping with our initial findings that GGTi inhibits IL-10 expression. Furthermore, addition of squalene did not rescue the IL-10 deficiency induced by atorvastatin, unlike mevalonate or GGPP. We thank the reviewer for this suggestion, and believe that this further strengthens the conclusion that geranylgeranylation is the specific mechanism by which cholesterol metabolism controls IL-10.

2. The data in Figure 5F show the ability of the authors to achieve knockdown of gene expression in human B cells. Given the central conclusion of the study regarding the role of GGTase and geranylgeranylation in TLR9-mediated IL10 induction, it would seem appropriate to show that this result is obtained in siRNA-mediated knockdown cells (and not just with chemical inhibitors).

We agree with the reviewer that this would be an ideal experiment to further confirm our findings. However, despite multiple attempts to do so, we have been unable to achieve siRNA mediated knockdown of GGTase as we did for BLIMP1. As the reviewer implies, it is notoriously difficult to achieve a successful knockdown in primary human B cells.

RevFig 3. Viability of B cells after nucleofection. Human B cells were nucleofected (or not), and then stimulated at the indicated time point after nucleofection. Stimulation immediately after nucleofection results in good viability, but a delay results in severely compromised viability,

Interestingly, a major difference between GGTase and BLIMP1 knockdown is the inducible nature of each protein. As BLIMP1 is only induced after stimulation, this allowed us to stimulate cells shortly after nucleofection +/- siRNA treatment and likely provided a more permissive context for gene targeting. In contrast, GGTase is constitutively expressed, so it was necessary to knock the protein down some hours prior to stimulation. We found that by leaving B cells unstimulated 6-12 hours after nucleofection – in an attempt to knock down GGTase before TLR9 ligation – the conditions severely compromised cell viability (See RevFig 3). We trust that the use of a different GGTase inhibitor, together with the aforementioned rescue experiments, are sufficiently convincing to demonstrate evidence of specificity.

3. Since the mevalonate pathway has two main outcomes (cholesterol and isoprenoids), it would seem appropriate to demonstrate whether or not addition of a cholesterol-committed metabolite such as squalene to the statin treated cells has any effect on IL-10.

As mentioned above, we have conducted this experiment and added the data to the manuscript. This demonstrated a failure to rescue IL-10 expression after statin treatment, in agreement with our conclusions.

4. Given the crucial role of small G-proteins in many B cell activation events, one wonders if the selective effect observed in this study is the dependence of IL10 on such isoprenylated proteins or, rather, the lack of TNF dependence on them. The effect of GGTi treatment on various other B cell activation outputs (other cytokines, activation marker expression, proliferation, viability) needs to be shown.

As also addressed by reviewer 1, we have conducted further experiments to more robustly address these concerns. We have now included data showing proliferation, differentiation, viability, and antibody production in longer term (5-7 day) cultures. (text from above response to reviewer 1: To answer this point, we have analysed the contribution of cholesterol metabolism generally, and also geranylgeranylation specifically, to the above processes. Specifically, we cultured cells for 5-7 days after

TLR9 ligation in the presence of either atorvastatin or GGTi, and subsequently analysed proliferation, differentiation, viability (all 5 days), and antibody production (7 days). We found that incubation of cells with atorvastatin had no effect on either proliferation, differentiation, viability, or antibody production. Similarly, GGTi had no effect on viability or antibody production, and minimal effects on proliferation and differentiation. We therefore conclude that this isoprenylation event selectively drives the induction of a regulatory phenotype in B cells that is independent of their differentiation). This is included in Supplementary Figure 5c and referenced in the text on lines 135-137.

To further address this, we have undertaken additional analysis of the RNA sequencing experiment outlined in Figure 5a-d, conducted on control or GGTi-treated B cells after stimulation through TLR9. We have extracted expression levels for genes involved in a broad range of cellular processes using GO terms for antigen presentation, cytokine activity, and apoptosis (RevFig 4a). As illustrated by the heatmaps (note that these data are not filtered for expression level cut-off, fold change, or FDR, but instead are normalised gene counts to give an impression of the global pathway), we see no global alterations, with some genes trending up and some trending down (RevFig 4A). This suggests that these fundamental cellular processes are not adversely affected by perturbations of GGTase activity. Interestingly, we see a consistent increase in IL1B expression in cells treated with GGTi. This would be consistent with data from MKD patients suggesting that increased IL-1b expression contributes to inflammatory disease activity. We have also extracted expression levels for transcription factors that are

RevFig 4. General transcriptional profile of control or GGTi-treated human B cells. **A.** Global expression of genes involved in the highlighted GO terms. Data shown in the heatmap are normalized counts with no statistical analysis applied, to highlight the global profile of these pathways. **B.** Selected transcription factors important to B cell activation and differentiation.

critical for defining B cell activation and differentiation (e.g. EBF1, BACH2, MYC), and again see no major defect in expression levels of these genes (RevFig 4B). Coupled with our longer-term cultures, we see no major defect B cell biology, which suggests to us that geranylgeranylation has a specific effect on the regulatory feature of B cells. Given that we have included data on longer term cultures in the revised manuscript regarding proliferation, differentiation, and viability, we have not included the data in RevFig4 in the manuscript at this point, but would be happy to do so if the reviewer felt that this adds important additional information to support our conclusions.

5. The authors state: "Isoprenyl modifications almost exclusively regulate the localization of Ras superfamily proteins", and so investigated selected Ras family signaling pathways downstream of TLR9. However, the cited review does not make this statement and the quantitatively dominant targets of geranylgeranylation are still poorly defined.

Thank you for highlighting this, and we have now altered this statement to, "The most well characterized isoprenylation targets are Ras superfamily proteins" on line 152. We have also included a relevant reference from Maurer-Stroh et al. Plos Biology, 2007 that aims at understanding isoprenylation targets at a global level through computational prediction.

6. It is stated: "Gene set enrichment analysis identified defective Ras (KRAS) and PI3K-Akt-mTOR signaling pathways in MKD patients when compared to healthy donors, in line with our own observations (Figure 6F)". However, KRAS is farnesylated not geranylgeranylated according to the cited review (it can be geranylgeranylated, but usually when FTase is first inhibited or GGTase1 is overexpressed (cross-prenylation)). Also, when interrogating these data, the ERK signaling shown earlier to have an effect on IL-10 is not mentioned.

As the reviewer mentioned above, the targets of geranylgeranylation are incompletely defined. There are discrepancies in the literature, with a report suggesting that KRAS interactions with PI3K are mediated by the geranylgeranyl modifications in macrophages (Akula, MK, Nat Immunol, 2016). We also report that bioinformatic predictions of isoprenyl modifications suggest that KRAS is a likely candidate for GGPP tagging (see RevTable 1). Whilst we did not intend to suggest that KRAS is the primary mediator of our phenotype in human B cells, we sought to highlight that MKD patients appear to

	GGTase			FTase		
	Prediction	Score	Pval	Prediction	Score	Pval
KRAS	+++	3.26	0.0003	++	1.44	0.0003
HRAS	--	-7.2	0.069	++	1.31	0.0004
NRAS	++	1.22	0.0017	++	0.34	0.0036
RRAS2	++	1.27	0.0016	++	1.21	0.0006

RevTable 1. Predictions of isoprenylation from Prenylation Prediction Suite. Bioinformatic analysis tool created by Maurer-Stroh and Eisenhaber, Genome Biol, 2005, predicting likely isoprenylation modifications based on protein sequence motifs. Data shown are Ras subfamily proteins expressed in resting B cells.

possess defective PI3K and Ras signalling events when analysed through an unbiased GSEA analysis, which would be in broad agreement with our findings. pERK did not show significant enrichment in the GSEA analysis, and in line with this, we saw a less pronounced inhibition of pERK upon GGTi treatment, as compared to pAkt. We hypothesise that MKD patients may also show a less pronounced pERK impairment and resulting expression signature, and therefore would not be highlighted by GSEA.

7. It is stated that addition of GGPP rescues the defective IL10 production in CpG stimulated MKD B cells. However, the rescue does not appear to be statistically significant (Fig. 6C). While it is recognized that the patients are rare, it would have been more compelling if additional data (such as technical repeats) were available. Comparison to the effect of squalene addition would also have been informative.

The statistical test that we used compared HD versus MKD and HD versus MKD+GGPP, and not MKD versus MKD+GGPP, as we asked if GGPP supplementation in MKD B cells meant that IL-10 production was equivalent to that seen in HD B cells. This showed that there was no significant difference in HD vs MKD+GGPP. Taking the reviewer's comment on board, we have revised our statistical analysis, instead comparing: HD versus MKD, HD versus MKD+GGPP (both unpaired t test), and MKD versus MKD+GGPP (paired t test as the donor is the same +/- GGPP). We can therefore now state that GGPP supplementation significantly increases IL-10 expression in MKD B cells. In order to show biological versus technical repeats with MKD patients, we have changed the data visualisation in Figure 6 and Supplementary Figure 8. This also allows a better comparison between matched healthy donors and MKD patients, where the matched experiments are conducted on the same day. Nonetheless, please see the revised statistical tests attached below (RevFig 5). We thank the reviewer for pointing this out.

We appreciate the helpful comments regarding further experiments in MKD patients, and have now added these. After a little delay, we were able to acquire samples from two new MKD patients, and have analysed the impact of squalene supplementation, and as the reviewer suggested, included technical repeats in one of the donors where cell

RevFig 5. IL-10 expression in HD and MKD patients. IL-10 expression in B cells from healthy donors (HD) or mevalonate kinase deficient patients (MKD) stimulated through TLR9 +/- geranylgeranyl pyrophosphate (GGPP) supplementation

numbers permitted. In contrast to GGPP, squalene was unable to alter the expression of IL-10 in MKD patients, suggesting that indeed a lack of GGPP synthesis is specifically the cause of defective IL-10 production. These data are now included in Supplementary Figure 8e and referenced in the text on lines 239-242.

To further characterise these patients with respect to our findings in healthy donors, we investigated if GSK3 inhibition would also be able to rescue IL-10 expression in MKD patients. Indeed, like GGPP, GSK3 inhibition was able to rescue IL-10 expression in MKD patients (these data are now included in Figure 6e and referenced in the text on lines 236-239 and lines 362-364). We feel this data links nicely to our previous findings that GSK3 blockade is able to rescue IL-10 expression after GGTase inhibition (Figure 3g). Finally, we also attempted to link our data regarding BLIMP1-mediated control of IL-10. Surprisingly, we observed that MKD patients show lower levels of BLIMP1 expression after TLR9 ligation, despite having greater frequencies of memory B cells, which we hypothesise may explain the reduced IL-10 expression (this data is now included in Figure 6, and discussed in the text on lines 365-369).

Minor

1. "Cholesterol metabolism" as referenced in the title seems to vague given that the paper focuses on GGPP (and protein prenylation) rather than cholesterol. A more appropriate title might be: "GGPP is required for IL-10 production by regulatory B-cells"

We agree with the reviewer that the title could be made more clear. We have therefore amended the title to, "Cholesterol metabolism drives regulatory B cell IL-10 through provision of geranylgeranyl pyrophosphate".

2. The name of the FTi inhibitor and the ERK inhibitor needs to be provided. The names of the inhibitors should be provided in the main text or figure legends and not just in the methods.

We have now provided the inhibitor names in the methods and also added these to each figure or figure legend.

3. The focus on PI3K-AKT as one of the downstream pathways is reasonable but this is quite possibly just one of many downstream pathways involved. It is important not to suggest that there is a selective role of geranylgeranylation in the activation of this pathway. For example, the data show that there is a reduction in pERK as well as pAKT.

We have added text to the discussion on this point on lines 322-323.

4. The differences between these findings for B cell IL10 expression and the earlier findings of cholesterol biosynthesis requirements in T cell IL10 production (Nat Commun 10, 498) is not adequately discussed.

We have now added further text in the discussion relating to our previous manuscript on lines 282-284.

Reviewers' comments:

Reviewer #1 (Remarks to the Author):

Review of NCOMMS-19-37939A

Bibby et al have addressed some of my concerns and the manuscript has improved accordingly. However, I still have considerable problems with the concept that all B cell populations contribute to IL-10 production and do so via Blimp1. Conceptionally, this just does not make sense to me. I am prepared to believe that B cells produce IL-10 in a GGPP-dependent manner. Similarly, I am fundamentally convinced that Blimp1 regulates IL-10 production in B cells. The conceptional problem I have is that the authors claim that the GGPP and Blimp1 pathway are linked (for which I don't see evidence) and that it applies to all B cell populations.

For example, naïve B cells don't express Blimp1, but according to the authors express IL-10. Of course, a lot may depend on how one defines 'naïve B cells'. After naïve B cells have been stimulated for 48h they may well express Blimp1 and IL-10. But after 48h stimulation, naïve B cells are not naïve B cells anymore, they are activated B cells. This is not just semantics.

Some of these interpretations hinge on the tSNE plot in Suppl. Fig. 4. It shows essentially no resolution of the various B cell populations. There is just one large cluster and an additional small cluster, neither appears to correlate to any marker used in the analysis. Neither naïve B cells, nor memory, B10 or plasmablast constitute a discrete population in the tSNE plot. How can that be?

One way of addressing the ongoing issue of which populations express IL10 under which conditions would be to sort human B cell populations into the relevant populations before culture. In this case, it would also be useful to CTV label the cells to allow measuring whether or not cell division parallels the acquisitions of IL-10 production capacity.

In this regard, it would also be useful to do these experiments with other ligands that induce B cell activation and differentiation. What about LPS and CD40L plus cytokine stimulation? Do these stimuli induce IL-10 in a GGPP-dependent manner? Is IL-10 production from MKD patients impaired?

A technical concern: after 48h of stimulation memory B cells should differentiate into plasma blasts. Based on their FACS analysis, this does not appear to happen in the cultures of the authors. Is that because they gate on CD20+ cells? It should be noted that CD20 is downregulated on plasmablasts. Thus, gating should not use CD20high.

Reviewer #2 (Remarks to the Author):

The authors have adequately addressed my concerns in their revised manuscript.

We thank the reviewer for their helpful suggestions. We were encouraged to read that they felt the manuscript had improved. We have made additional textual changes to the manuscript, and provided new data to address the specific points raised. Please find below a point by point response to the comments of the reviewer. All textual changes are highlighted in orange in the revised manuscript.

Response to reviewer

Bibby et al have addressed some of my concerns and the manuscript has improved accordingly. However, I still have considerable problems with the concept that all B cell populations contribute to IL-10 production and do so via Blimp1. Conceptionally, this just does not make sense to me. I am prepared to believe that B cells produce IL-10 in a GGPP-dependent manner. Similarly, I am fundamentally convinced that Blimp1 regulates IL-10 production in B cells. The conceptional problem I have is that the authors claim that the GGPP and Blimp1 pathway are linked (for which I don't see evidence) and that it applies to all B cell populations.

We provide direct evidence of BLIMP1 regulation by GGPP, as inhibition of its cognate enzyme, GGTase, prevents the inducible expression of BLIMP1 (Figure 5a-d). In line with this, targeting GGTase or BLIMP1 attenuates IL-10 production, providing more evidence for the link between GGPP, BLIMP1, and IL-10. These data are presented in Figure 5b-i, and Supplementary Figure 7c. These data show that the GGPP-dependent proximal signalling events occurring upon TLR9 ligation regulate downstream expression of BLIMP1, whose expression is required for IL-10 production. This was subsequently supported by data from MKD patients who demonstrate a poor IL-10 response due to insufficient GGPP synthesis, and reduced BLIMP1 upregulation upon CpG stimulation.

For example, naïve B cells don't express Blimp1, but according to the authors express IL-10. Of course, a lot may depend on how one defines 'naïve B cells'. After naïve B cells have been stimulated for 48h they may well express Blimp1 and IL-10. But after 48h stimulation, naïve B cells are not naïve B cells anymore, they are activated B cells. This is not just semantics.

Following well established literature, we have defined naïve B cells as IgD+CD27-, and memory B cells as IgD-CD27+. Using this traditional definition, we have shown that recently activated naïve B cells do indeed express BLIMP1 and IL-10 after CpG stimulation, albeit at lower levels than memory B cells (Figure 6F, and Supplementary Figure 7F; also see Response Figure 1 which includes BLIMP1 expression in unstimulated versus stimulated B cells including an additional fluorescence minus one control). This was an interesting and surprising finding to us too, as we were unaware of any literature assessing the capacity of naïve B cells to express BLIMP1 upon CpG stimulation. These cells may well go on to differentiate into memory B cells, but they are not currently memory B cells based on traditional phenotyping. Neither is their ability to

Response figure 1. BLIMP1 expression in B cell subsets. B cells were either left unstimulated or stimulated with CpG for 40 hours. Cells were then harvested and stained for BLIMP1 alongside IgD and CD27. Also included is a fluorescent minus one control showing a 'true negative' for BLIMP1. As expected, we see little/no BLIMP1 expression in unstimulated IgD+CD27- (naïve) cells. However, cells still possessing a naïve phenotype (IgD+CD27-) after 40 hours do express BLIMP1, which as expected, is lower than that seen in memory (IgD-CD27+) B cells. We propose calling these phenotypically naïve cells that have received CpG stimulation 'recently activated naïve B cells'.

produce IL-10 dependent on their differentiation stage, as these IgD+CD27- cells clearly express IL-10. Therefore, as these cells still possess the phenotype of IgD+CD27- at 40 hours after stimulation, we have maintained their classical definition as naïve B cells, whether or not they may have become recently activated. We appreciate the reviewer's concern on this point, and so have therefore altered the text to reflect this. In the text we have now referred to these cells as 'recently activated naïve B cells', recognising that these are not completely antigen naïve (lines 206, 347-348, highlighted CpG activation on line 110, and highlighted CpG stimulation prior to naïve B cell classification in Supplementary Figure 4a in the revised manuscript). We trust that this change will provide additional clarity as to the exact definition of these IgD+CD27-BLIMP1^{low} cells. Please also see Response Figure 1, which highlights this textual change in the context of BLIMP1 expression analysis.

Some of these interpretations hinge on the tSNE plot in Suppl. Fig. 4. It shows essentially no resolution of the various B cell populations. There is just one large cluster and an additional small cluster, neither appears to correlate to any marker used in the analysis. Neither naïve B cells, nor memory, B10 or plasmablast constitute a discrete population in the tSNE plot. How can that be?

We apologise if the presentation of the tSNE analysis has been unclear, recognising that the lack of distinct clusters within the total B cell data structure (referring to panel 1 in Response Figure 2) after tSNE analysis could be confusing. However, the presence of distinct clusters should not be expected. To outline why, below is a detailed explanation of the tSNE analysis. We have also included a proposal for an improved visualisation using the same data, which we have now included in Supplementary

Response figure 2. tSNE analysis of human B cells after TLR9 stimulation, using the same data from Supplementary Figure 4a. B cells were stimulated with CpG for 40 hours, and stained for markers: CD20, CD24, CD27, CD38, IgD, and IL-10. The parameters used for tSNE analysis were: iterations = 1000, perplexity = 100, learning rate = 200. Here highlighted are total B cells (first plot), naïve vs memory B cells within total B cells (second plot), CD24^{hi}CD38^{hi} and B10 cells within total B cells (third plot), and IL-10+ B cells within total B cells (fourth plot).

Figure 4a, replacing the current tSNE plots. We feel this addresses the reviewer's concerns about the ability to adequately resolve B cell populations.

We provided the tSNE data as a supplementary figure as it is in complete agreement with current literature and supports the data in our manuscript. We chose tSNE analysis as this provided an unsupervised method (not possible by conventional flow cytometry gating) to assess where B cell populations align relative to each other, and if IL-10 was a distinguishing feature of any particular phenotype. The two main points that the tSNE analysis demonstrated were (1) B cell populations do cluster in their classical populations (naïve, memory, B10, and CD24^{hi}CD38^{hi}), with B10 showing an overlap with memory, and CD24^{hi}CD38^{hi} overlapping with naïve, as expected, and (2) the IL-10 signal is a standout marker that is not isolated to any particular B cell phenotype, as it clusters separately, and includes cells from all B cell populations; in keeping with the phenotyping data throughout.

In an attempt to better clarify this, please see Response Figure 2 for a revised presentation of the same data; this includes a slight modification of the tSNE parameters to improve cluster resolution. This aims to highlight the independent clustering of cell populations by plotting the total data structure separately (panel 1 of Response Figure 2), and then combining naïve and memory cells (panel 2), B10 and CD24^{hi}CD38^{hi} cells (panel 3), and IL-10+ cells (panel 4). This is in contrast to the individual plots we provided in Supplementary Figure 4a of the previous manuscript version.

Reassuringly, the tSNE analysis shows clear segregation of mutually exclusive B cell populations within the tSNE plot i.e. we see naïve and memory B cell populations in independent zones of the tSNE plot (See panel 4-5 of Supplementary Figure 4a, and panel 2 in Response Figure 2). We would not necessarily expect to see these form completely separate cluster 'islands' in the total B cell data structure (outlined in gray in Supplementary Figure 4a, and panel 1 in Response Figure 2), as this depends on multiple technical aspects behind the dimensionality reduction in the tSNE algorithm

including: the number of markers included, the shared markers between populations despite their exclusive phenotype (e.g. CD20, IL-10, and CD24), and the number of iterations, perplexity, and learning rate used in the tSNE analysis. For example, the similarity of naïve and memory B cells both expressing high levels of CD20 would prevent completely segregated cluster islands forming within the tSNE plot, as this similarity may be weighted more highly than the difference in IgD/CD27 expression (for an example of B cell tSNE analysis showing similar data structure, please see Figure 1h of Jellusova et al., Nat Immunol, 2017). It is worth highlighting, in this context, that the reason completely separate cluster islands are so commonly seen in mass cytometry or single cell RNA-sequencing tSNE analyses (in comparison to flow cytometry used here) is that when many more markers are combined, it allows the tSNE algorithm to more clearly 'see' the distinctions between populations.

The reason that CD24^{hi}CD38^{hi} and B10 populations do not possess completely discrete tSNE zones is because these B cell populations are contained within naïve and memory B cell populations respectively. For example, CD24^{hi}CD38^{hi} B cell populations are a population of naïve B cells, and therefore occupy a zone of the tSNE plot within the naïve B cell zone. Similarly B10 cells are considered a mature B cell population expressing CD27, and this population therefore overlaps with the memory B cell population in the tSNE analysis (comparing panels 2-3 in Response Figure 2). Indeed, it would be alarming if CD24^{hi}CD38^{hi} and B10 cells did occupy discrete population clusters. Moreover, it was interesting that IL-10 expression generated a cluster of its own. This fits well with our observation that IL-10 is not produced by a sole subset (i.e. naïve, memory, B10, or CD24^{hi}CD38^{hi}), but stands out as a distinguishing subset feature of its own.

Consequently, the overlapping of memory-B10 and naïve-CD24^{hi}CD38^{hi}, and non-overlapping of naïve-memory populations validates our tSNE analysis. Exchanging this Response Figure 2 with Supplementary Figure 4a does not require any textual change to the main manuscript, but we have fully explained this in the figure legend and methods of the revised manuscript (lines 491-496). We hope that this better explains the key messages that we are trying to communicate.

One way of addressing the ongoing issue of which populations express IL10 under which conditions would be to sort human B cell populations into the relevant populations before culture. In this case, it would also be useful to CTV label the cells to allow measuring whether or not cell division parallels the acquisitions of IL-10 production capacity. In this regard, it would also be useful to do these experiments with other ligands that induce B cell activation and differentiation. What about LPS and CD40L plus cytokine stimulation? Do these stimuli induce IL-10 in a GGPP-dependent manner? Is IL-10 production from MKD patients impaired?

Response figure 3. IL-10 expression in relation to B cell proliferation. IL-10 expression in human B cells stimulated with CpG for 5 days, relative to proliferation stage determined by CellTrace Violet.

Here we provide additional data, albeit conducted using bulk B cells, that looks at the capacity of B cells to express IL-10 in the context of proliferation. We see no evidence to suggest that expression of IL-10 is in any way related to the capacity of B cells to proliferate, as determined by co-staining for CTV and IL-10. The data confirm that earlier generations, including undivided cells, express comparable levels of IL-10 when compared to later generations (Response Fig 3).

The reviewer asks about assessing the GGPP dependency for other receptor signalling pathways that induce IL-10 in human B cells. We have already conducted these experiments but did not include the data, as they were not so informative. Therefore, we now provide data assessing the response of human B cells to other such ligands mentioned above, including recombinant CD40L, LPS, and IFN α . In general agreement

Response figure 4. TLR9 ligand CpG versus other stimuli for IL-10 production in human B cells. Total CD19+ B cells were left unstimulated, or stimulated with CpG (TLR9), CD40L (CD40), LPS (TLR4), IFN α , or a combination, with all stimulations indicated below the histograms. Data are presented as mean \pm SD, with points indicating individual donors. Statistical analysis was conducted using a Friedman's test followed by a Dunn's multiple comparisons test comparing all conditions to unstimulated cells; *P<0.05, **P<0.01

with the literature, we see that CpG is the most potent single stimuli for the induction of IL-10. In comparison, we see little IL-10 produced with other stimuli alone, and no further enhancement when used in combination with CpG (Response Figure 4). We therefore were not in a position to test the GGPP dependency for IL-10 in response to these specific stimuli, and consequently, given the limited samples, did not go on to address this in MKD patients. Other groups have established protocols for the induction of IL-10 by CD40L, but these typically involve irradiated CD40L expressing cell lines (Blair, P. et al, Immunity, 2010). Our use of recombinant CD40L could potentially explain discrepancies with these data. With regards to LPS, resting human B cells – unlike murine B cells – express little or no TLR4, which is likely the reason why we see no IL-10 production with LPS. Furthermore, IFN α has previously been suggested to increase B cell IL-10 when supplemented alongside TLR7, but not TLR9 activation (Liu, BS, et al. Eur J Immunol, 2014).

With regards to the proposal for sorting B cells. We have demonstrated in the manuscript and rebuttal letters that all B cell populations express IL-10 (Supplementary Figure 4a and c). In addition, we have demonstrated that all B cell populations upregulate BLIMP1 in response to CpG, albeit to varying degrees (Figure 5e-f, Figure 6f, Supplementary Figure 7e-f). Further, we demonstrated that BLIMP1 is required for IL-10 expression (Figure 5g-i), and BLIMP1 expression itself is dependent on GGPP regulated TLR9 signalling cascades (Figure 5b-d). With the data included here in Response Figure 3, we can also show that the ability to express IL-10 does not parallel the proliferative state of B cells. This would strongly agree with data from MKD patients, whereby B cells do not possess a proliferative or differentiation defect (Supplementary Figure 8c), but do show a defect in IL-10 expression across all B cell populations (Figure 6c-d), and poor upregulation of BLIMP1 in response to TLR9 ligation, again, across all B cell populations (Figure 6f). Therefore, we think that the data presented provides sufficient evidence of a common mechanism across cell subsets. As phenotyping expression of IL-10 across B cell subsets only accounts for a minimal portion of this work, and our main findings are uncovering the mechanistic control (defining the cholesterol metabolite, delineating signalling pathways, and transcriptional control of IL-10) of IL-10 by geranylgeranylation in healthy B cells and MKD patients, we do not feel that further characterising B cell subsets by performing additional sorting experiments would significantly advance or alter the overall message we are trying to convey.

A technical concern: after 48h of stimulation memory B cells should differentiate into plasma blasts. Based on their FACS analysis, this does not appear to happen in the cultures of the authors. Is that because they gate on CD20+ cells? It should be noted that CD20 is downregulated on plasmablasts. Thus, gating should not use CD20high.

We were aware of this, and therefore did not use CD20 as an exclusion gate. Our gating strategy was: Live/dead exclusion > Gating on cells > Gating on single cells > these cells then used for analysis (see Response Fig 5). We are happy to include this as

a supplementary figure if the reviewer feels this adds clarity. Most of our cultures were conducted for 40 hours. At this time point, we see little generation of plasmablasts, as this is likely not long enough for full differentiation. However, at 5 days, we see significant accumulation of plasmablasts, as expected (Response Figure 5). We hope that the reviewer finds this reassuring with respect to the technical validity of our cultures in terms of B cell differentiation, and the gating used in the analysis.

Response figure 5. General gating strategy used in the analysis of human B cells after culture

Response figure 6. Formation of plasmablasts over time in response to CpG. Total B cells were stimulated, or not, with CpG and cultured for the indicated time, subsequently gating for plasmablasts using CD27⁺CD38^{hi}

Reviewer #1 (Remarks to the Author):

Bibby et al have provided extensive explanation for their data and model. It is clear that there is in some ways a fundamentally different understanding of some of the data. A good example is the tSNE plot in Suppl. Fig. 4. The authors interpret these results as supportive of their argument that IL-10 producing cells do not constitute a separate population. I would argue it show exactly that: IL-10+ B cells sit separate from all other cells and do not or hardly overlap with any of the other populations. Thus, our interpretations of the data differ quiet fundamentally.

I also argue that the current way of showing the data to some degree defies the purpose of tSNE plots. After clustering, these plots should display the expression levels of certain markers, such as IL-10, CD24, CD38 etc within these plots. That would allow readers to assess the data. At present, the authors gate populations manually and then overlay them onto their tSNE plot. This is not helpful, and I would suggest to change that.

I am also not convinced of the data that show induction of Blimp1 by GGPP (Fig. 5). The only evidence is the RNAseq data; however, this should have been confirmed by an independent method.

Having said all of that, I am prepared to accept that the study is fundamentally interesting and provides an expansion of the model as to how IL-10 production from B cells is regulated. As such, it will be of interest to the community and will undoubtedly stimulate discussion and follow-up studies.

I suggest to include most of the data and figures provided to the reviewer as Suppl. Data.

Response to reviewer

Thank you again to the reviewer for reviewing the manuscript. Below, we provide a more detailed response to the specific concerns raised, including an additional figure.

Reviewer #1 (Remarks to the Author):

Bibby et al have provided extensive explanation for their data and model. It is clear that there is in some ways a fundamentally different understanding of some of the data. A good example is the tSNE plot in Suppl. Fig. 4. The authors interpret these results as supportive of their argument that IL-10 producing cells do not constitute a separate population. I would argue it show exactly that: IL-10+ B cells sit separate from all other cells and do not or hardly overlap with any of the other populations. Thus, our interpretations of the data differ quiet fundamentally.

I also argue that the current way of showing the data to some degree defies the purpose of tSNE plots. After clustering, these plots should display the expression levels of certain markers, such as IL-10, CD24, CD38 etc within these plots. That would allow readers to assess the data. At present, the authors gate populations manually and then overlay them onto their tSNE plot. This is not helpful, and I would suggest to change that.

Since the tSNE analysis has caused confusion, we think it best to omit it from the paper. We have already included a supervised FACS gating in supplementary Figure 4c that demonstrates our point sufficiently. This shows that IL-10 is produced from multiple B cell populations spanning naïve to memory B cells.

In a brief response, the reviewer expresses concerns about our interpretation, namely, "The authors interpret these results as supportive of their argument that IL-10 producing cells do not constitute a separate population". We only mean to suggest that IL-10 is the defining marker of 'regulatory' B cells, and the expression of any specific surface marker does not encompass all B cells with a regulatory capacity. Therefore, cells do separate in the tSNE plot based on IL-10, but IL-10 is not restricted to any particular phenotype, as all other populations are in this IL-10 cluster. This is in agreement with the majority of literature on IL-10 producing B cells, which concludes that there is no defining phenotype of IL-10 expressing B cells, besides IL-10 itself. Nonetheless, as mentioned above, the we share the view that the tSNE analysis may cause confusion, so we are happy to take this out altogether. This will not affect our conclusions in the manuscript.

I am also not convinced of the data that show induction of Blimp1 by GGPP (Fig. 5). The only evidence is the RNAseq data; however, this should have been confirmed by an independent method.

We agree with and thank the reviewer for raising this, as we missed this when putting the manuscript together. We can now include the data below from 2 independent donors (also different donors than those used in the RNA-seq), showing protein expression of BLIMP1 after stimulation in the presence or absence of the GPP inhibitor, GGTi. In agreement with the RNA-seq data, we see around a 50% reduction in BLIMP1 expression in GGTi treated cells when compared to cells in the absence of the inhibitor. We have now included this (Sup Fig 1) in Supplementary Figure 7e.

Sup Fig 1. BLIMP1 expression is downregulated after GGTi treatment. Human B cells were stimulated with CpG in the presence or absence of GGTi for 40 hours, and then protein was extracted for analysis by western blotting. Data show 2 independent donors.

Having said all of that, I am prepared to accept that the study is fundamentally interesting and provides an expansion of the model as to how IL-10 production from B cells is regulated. As such, it will be of interest to the community and will undoubtedly stimulate discussion and follow-up studies.

I suggest to include most of the data and figures provided to the reviewer as Suppl. Data.

With regards to inclusion of figures, the following list of figures generated during the reviewer correspondence are now included in the manuscript:

Figure 6e-f

Supplementary Figures 2a-b

Supplementary Figure 4c

Supplementary Figures 5b-f

Supplementary Figure 6c

Supplementary Figures 7e-g

Supplementary Figure 8c and 8e

and

Removal of Supplementary Figure 4a (tSNE analysis)